# Towards Next-Generation Logic Synthesis:
# A Scalable Neural Circuit Generation Framework

**Zhihai Wang**[1][*]    **Jie Wang**[1][†]    **Qingyue Yang**[1]    **Yinqi Bai**[1]    **Xing Li**[2]    **Lei Chen**[2]

**Jianye Hao**[2,3]    **Mingxuan Yuan**[2]    **Bin Li**[1]    **Yongdong Zhang**[1]    **Feng Wu**[1]

[1]MoE Key Laboratory of Brain-inspired Intelligent Perception and Cognition,
University of Science and Technology of China
[2]Noah's Ark Lab, Huawei Technologies
[3]College of Intelligence and Computing, Tianjin University

## Abstract

Logic Synthesis (LS) aims to generate an optimized logic circuit satisfying a given functionality, which generally consists of circuit translation and optimization. It is a challenging and fundamental combinatorial optimization problem in integrated circuit design. Traditional LS approaches rely on manually designed heuristics to tackle the LS task, while machine learning recently offers a promising approach towards next-generation logic synthesis by *neural* circuit *generation* and *optimization*. In this paper, we first revisit the application of differentiable neural architecture search (DNAS) methods to *circuit generation* and found from extensive experiments that existing DNAS methods struggle to exactly generate circuits, scale poorly to large circuits, and exhibit high sensitivity to hyper-parameters. Then we provide three major insights for these challenges from extensive empirical analysis: 1) DNAS tends to overfit to too many skip-connections, consequently wasting a significant portion of the network's expressive capabilities; 2) DNAS suffers from the structure bias between the network architecture and the circuit inherent structure, leading to inefficient search; 3) the learning difficulty of different input-output examples varies significantly, leading to severely imbalanced learning. To address these challenges in a systematic way, we propose a novel regularized triangle-shaped circuit network generation framework, which leverages our key insights for *completely accurate* and *scalable* circuit generation. Furthermore, we propose an evolutionary algorithm assisted by reinforcement learning agent restarting technique for efficient and effective neural *circuit optimization*. Extensive experiments on four different circuit benchmarks demonstrate that our method can precisely generate circuits with up to 1200 nodes. Moreover, our synthesized circuits significantly outperform the state-of-the-art results from several competitive winners in IWLS 2022 and 2023 competitions.

## 1   Introduction

Complex integrated circuits (ICs) can have billions of transistors, making purely human-based design impossible [1]. To tackle this problem, the IC industry relies on electronic design automation (EDA) tools [2] that progressively transform a high-level hardware design into a layout ready for IC fabrication. Logic synthesis (LS) is a fundamental step in EDA which aims to transform a behavioral-level description of a design into an optimized gate-level circuit to minimize its delay and area. As

---

[*]This work was done when Zhihai Wang was an intern at Huawei.

[†]Corresponding author. Email: jiewangx@ustc.edu.cn.

38th Conference on Neural Information Processing Systems (NeurIPS 2024).

LS is the first step in EDA tool-chains that yields the final IC layout, the quality of its output highly impacts the area, power, and performance of the final ICs [3, 4].

LS is a challenging $\mathcal{NP}$-hard combinatorial optimization problem. Commercial and academic LS tools use sophisticated human-designed heuristics to approximately solve this task, often obtaining sub-optimal solutions. The synthesis of high-level designs to circuits is typically done as a direct translation of hardware description language code coupled with post-processing optimization. Recent works [5–7] have shown that there exists room for fusing these two steps with neural compiler architectures. Therefore, leveraging machine learning for direct *neural circuit generation and optimization* emerges as a significant direction towards next-generation LS.

In this paper, we first revisit the application of differentiable neural architecture search (DNAS) methods to synthesize circuits from input-output examples [5, 8, 7], which seems to offer a promising avenue towards neural circuit generation. Unfortunately, we found from extensive experiments that the existing method not only struggles to generate circuits precisely, particularly in large-scale circuits but also exhibits high sensitivity to hyperparameters. Through comprehensive empirical analysis, we summarize three key insights for these challenges. 1) DNAS suffers from the curse of skip-connections, tending to learn to too many skip-connections, which results in low utilization of the large network. 2) there is a discrepancy between DNAS and the inherent structure of circuits, leading to redundant search space and inefficient search. 3) the learning difficulties vary greatly across different input-output examples, resulting in a severe imbalance during the training process.

To address these challenges in a systematic way, we propose a novel regularized triangle-shaped circuit network generation framework, namely T-Net, which leverages our key insights for *completely accurate* and *scalable* circuit generation. To further enhance our T-Net for logic synthesis, we propose an evolutionary algorithm assisted by reinforcement learning agent restarting technique for further efficient and effective neural *circuit optimization*. The efficient search and scalable circuit generation of our T-Net come from the following aspects. 1) **Multi-Label Transformation of Training Data**. To enhance the scalability, T-Net proposes to partition the input-output examples into several sub-datasets based on the Shannon decomposition theorem and merge these sub-datasets to transform the original single-label data into multi-label data with significantly reduced input data. Jointly learning circuit structures of transformed input-output examples also exploits inherent circuit functionality symmetry for logic sharing and reducing generated circuit size. 2) **Triangle-Shaped Network Architecture**. Based on the key observation that the circuit structure generally follows a triangle-shape, T-Net designs a Triangle-shaped network architecture, which significantly reduces the search space, instead of common square-shaped architectures. 3) **Regularized Training Loss**. To mitigate overfitting to many skip-connections, T-Net proposes an inner-architecture regularized loss to suppress excessive skip-connections. Moreover, T-Net further proposes a hardness-aware loss function to actively optimize hard input-output examples.

We conducted extensive experiments on 18 circuits from four benchmarks. For circuit generation, our T-Net accurately generates large circuits with up to 1200 nodes, surpassing the state-of-the-art (SOTA) DNAS methods[5, 9], while also producing much smaller circuits compared to traditional methods[10, 11]. Based on our generated compact circuits, our evolutionary algorithm further optimizes circuits, significantly outperforming not only traditional methods, but also SOTA approaches from several competitive winners in IWLS 2022 and 2023 competitions.

We summarize our major contributions as follows. 1) An extensive analysis of the challenges inherent in applying Differentiable Neural Architecture Search (DNAS) for neural circuit generation was conducted, leading to three key underlying insights. 2) Leveraging these key insights, we developed T-Net, a neural circuit generation framework enabling efficient search and scalable generation. 3) Experiments on 18 circuits show that our approach achieves a significant 68% improvement in circuit area over the traditional method, and a remarkable 5.36% improvement compared to the SOTA approach employed by the winners of the IWLS 2023 competition.

## 2 Related Work

**Machine Learning in Logic Synthesis** In recent years, integrating machine learning (ML) into chip design workflows has garnered significant attention [1, 12–14]. The investigation spans two main areas: ML embedded in LS and end-to-end LS using ML techniques. ML embedded in LS involves incorporating ML into specific LS stages to enhance efficiency and quality. Notable efforts include using ML to tune optimization flows[15–17], predict metrics [14], and improve

decision-making[18, 19] in LS methods. ML for end-to-end LS includes research exploring replacing traditional LS stages with ML[1]. Approaches range from language-based circuit description[20, 21] to circuit generation through searches [5, 7]. Notable methods include integrating real-valued logic with continuous parameterization and using differentiable neural architecture search (DNAS)[6]. Despite the promising advancements, existing end-to-end methods face challenges in scaling to large circuits and are sensitive to hyperparameters. In this paper, we rethink the traditional DNAS methods for LS and propose a novel regularized triangle-shaped circuit generation framework.

**IWLS Contest** The International Workshop on Logic & Synthesis (IWLS) [8] annually hosts a contest, with themes in 2022 and 2023 [22] focusing on LS from input-output examples (i.e., truth tables), scored based on circuit sizes (i.e., node number). Participated teams mainly employ traditional methods [10, 11, 23–26] for circuit synthesis. In 2023, Google DeepMind introduced a DNAS-based method [27], achieving first place. We replicated it as a baseline, conducting a detailed analysis and enhancing the DNAS-based generation method. For circuit optimization, various operator sequence optimization approaches have been proposed [28, 29]. The 2022 champion EPFL team utilized Bayesian optimization methods within an EA framework [30]. In contrast, we employed RL methods with strong search capabilities and introduced a restart strategy to mitigate local optima.

## 3 Background

**Logic Synthesis (LS) from IO Examples** In recent years, a promising direction that synthesizing circuits from IO examples has received increasing attention [27, 31, 32, 5, 7]. Specifically, researchers aims to use machine learning to generate a circuit given a truth table that describes the functionality of the circuit. Note that each line in the truth table is an input-output pair, which means that given the input to the circuit it will produce the corresponding output. For machine learning (ML) domain, researchers formulate the truth table as a training dataset consisting of many input-output pairs, and aim to use a ML model to generate circuits that completely fits the dataset.

**Circuit Graph Representation** Boolean Networks are widely-used discrete mathematical models with applications in various fields [33]. In these networks, nodes represent Boolean functions, and edges illustrate connections between them. Boolean functions map from an n-dimensional space $B^n$ to a 1-dimensional space $B$, where $B = \{0, 1\}$. In the LS stage, circuits are often depicted as **And-Inverter Graphs** (AIG), offering a concise representation of Boolean Networks. AIGs consist of constant, primary inputs (PIs), primary outputs (POs), and two-input And nodes. Inverter edges signify an inversion signal. The size of a circuit denotes the number of And nodes in the AIG, while the depth (level) signifies the longest path from a PI to a PO.

**Traditional DNAS for LS from IO Examples** Recent works [5, 7] propose to leverage DNAS methods for generating circuit graphs from IO examples, which shows a promising direction towards next-generation logic synthesis. Specifically, they formulate a neural network as a circuit graph (i.e., AIG), where each neuron represents a logic gate (And gate) and connections between neurons represent wires connecting these logic gates. For a parameterized neural network, the neurons are fixed as logic gates, and the connections between neurons are parameterized as learnable parameters. To enable differentiable training via gradient descent, they introduce continuous relaxation into discrete components in the neural network. First, the logical operations of logic gates (neurons) are translated into their differentiable counterparts. For instance, $a\,AND\,b$ is relaxed to $a \cdot b$, and $NOT\,a$ is relaxed to $1 - a$ [6]. Second, discrete network connections are parameterized, employing Gumbel-softmax [34] during forward propagation to continuousize and sample the connections between nodes, thus enabling optimization through gradient descent to find high-quality solutions.

## 4 Rethinking DNAS for Neural Circuit Generation

In this section, we first present motivating challenges in using DNAS for neural circuit synthesis from input-output examples in Section 4.1. Then, we present a deep understanding of these challenges in Section 4.2. We provide the detailed experimental setup in AppendixB.3.

### 4.1 Motivating Challenges

We present two fundamental challenges in neural circuit generation of existing DNAS[5]. First, DNAS struggles to generate circuits exactly from input-output examples, especially for large-scale circuits. Second, DNAS exhibits high sensitivity to hyperparameters.

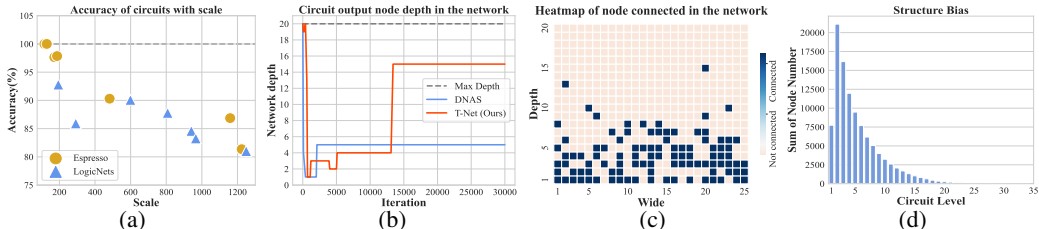

Figure 1: (a) DNAS struggles to accurately generate circuits, especially larger ones. (b) The depth of an output node of the circuit in the circuit. DNAS only connects to very shallow layers, while our method learns deeper layers. (c) The visualization of the converged DNAS network. The dark nodes represent the used circuit nodes, indicating very low utilization of deep-layer nodes. (d) The circuits generated by SOP show that the average number of nodes per layer forms a triangular pattern.

**Generating Exact Circuits is Challenging** To evaluate whether DNAS can generate circuits exactly, we evaluate the DNAS method as described in Appendix B.1 and C on the input-output examples from circuits in two benchmarks. Figure 1(a) shows that out of the 16 circuits, 14 can not be generated correctly. Therefore, it is extremely challenging for DNAS to generate functionally correct circuits.

Moreover, Figure 1(a) illustrates the relationship between circuit accuracy and circuit scale. The metric for circuit scale is quantified by the number of AIG nodes obtained through the traditional synthesis method. The results reveal nearly 20% degradation in circuit accuracy as the circuit scale increases. This demonstrates the challenge that DNAS confronts in generating accurate circuits as the circuit scale grows, highlighting the poor scalability of the DNAS method.

**Sensitivity to Hyperparameters** We evaluate robustness of DNAS in circuit generation using various initializations. Results showed up to a 14.5% accuracy variation depending on the random seed, highlighting the challenge of obtaining stable results with DNAS. More details in Appendix C.2.

## 4.2 A Deep Understanding of These Challenges

To elucidate the underlying causes of these challenges, we undertook comprehensive analytical experiments, which yielded the following three key insights. Firstly, DNAS suffers from the curse of skip-connections, tending to learn too many skip-connections, which results in low utilization of large initialization networks. Secondly, there is a discrepancy between DNAS and the inherent structure of circuits, leading to an inadequate exploration of the search space. Lastly, the varying learning difficulties among input-output examples cause an imbalance in the training process.

**The Curse of Skip-Connection** We observed that existing methods exhibit low utilization of the network when searching within a large network. This is attributed to the fact that connections can span across layers, bypassing certain nodes and excluding them from the final circuit. To investigate this, we analyzed how the skip connections evolve during training. The output nodes of a circuit are selected within the network through a set of learnable connection parameters. To study the cross-layer connection phenomenon of the outputs, we observe the depth of the output nodes within the network. Figure 1(b) shows the fluctuation in the depth of an output node during training. It is evident that the depth of the circuit output node undergoes a rapid decline to nearly 0, followed by a gradual rise and eventual stabilization at a shallow depth of 5. This observation implies that only a fraction of the network layers, specifically about a quarter, are effectively utilized in the circuit. The skip connections within the circuit span a considerable depth, significantly constraining the upper limit of the network's expressive capacity. As a point of contrast, our approach connects this output to layer 15, allowing for the full utilization of nodes.

Existing methods underutilize large networks because connections can span across layers, bypassing certain nodes and excluding them from the final circuit. To investigate this, we analyzed how skip connections evolve during training. We observed the depth of output nodes within the network to study cross-layer connections. Figure 1(b) depicts the depth fluctuation of an output node during training. The output node's depth initially drops sharply to almost 0, then gradually rises and stabilizes at a shallow depth of 5. This indicates that only about a quarter of the network layers are effectively utilized. Our approach connects this output to layer 15, enabling full node utilization.

To further illustrate the circuit structure searched by DNAS, we visualized the positions of circuit nodes within the network in Figure 1(c). Notably, this circuit has multi-outputs, resulting in layer

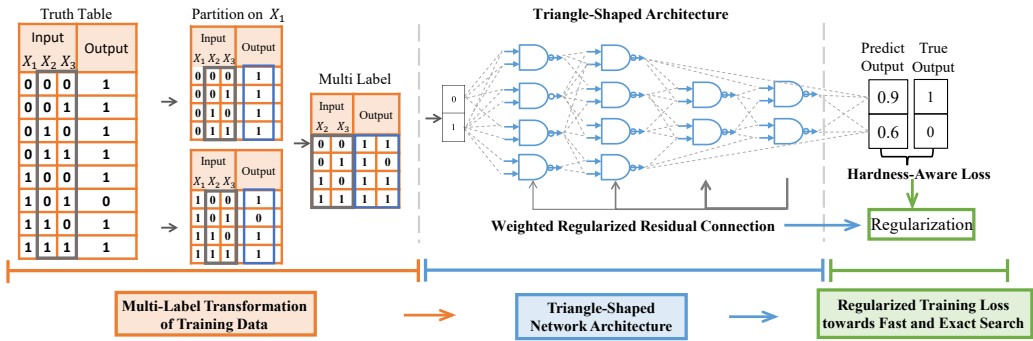

Figure 2: Framework of the Regularized Triangle-Shaped Circuit Network (T-Net). Our proposed T-Net consists of three key modules: 1) Multi-label transformation of training data to decrease generation difficulty. 2) A Triangle-shaped network architecture, designed to align with the structural biases inherent in logic circuits. 3) Regularized training loss for efficient search and exact generation.

configurations distinct from those in the single-output case shown in Figure 1(b). It can be observed that only a subset of bottom-layer nodes is integrated into the circuit, with about two-thirds of the nodes being left idle due to skip connections. This visual representation demonstrates that excessively distant skip connections diminish node utilization and the expressive capacity of the network.

We have noted that DNAS shows restricted exploration of the network during training. Specifically, a node is classified as *explored* if it is included in the discretized circuit at any point during training, and *exploration* is defined as the ratio of explored nodes to total nodes in the network. It is evident that exploration is more extensive in shallower layers, decreasing as the layers deepen, as illustrated in Appendix C. This finding prompts us to investigate whether structural bias within the circuit architecture is responsible for imbalanced exploration. Furthermore, we observed that the skip-layer count of the output bits is highly influenced by the hyperparameter random initialization, exhibiting significant fluctuations. This sensitivity highlights that the accuracy of circuit generation is extremely responsive to hyperparameters. During the training process, the occurrence of extensive skip-layer phenomena is attributed to the fact that choosing skip-connection leads to most rapid error decay during optimization. The network tends to learn skip-connection rather than traversing through more nodes. This is known as the curse of skip-connections, as mentioned in [35, 36].

**The Structure Bias of Circuits** To further investigate the structural bias in circuit design, we examine the structure of circuits generated by traditional methods. Utilizing Sum-of-Products (SOP) in ABC to synthesis circuits, we analyze and quantify the node distribution across different layers. Figure 1(d) presents the average node count distribution per layer in circuits. This reveals a distinct structural pattern: the circuits are wider in the bottom layers and become narrower in the deeper layers, suggesting inherent structural preferences in circuit designs. This is inconsistent with the commonly used rectangular network shape. Utilizing a rectangular network to learn circuit structures may result in a vast, redundant search space in the deep layers, leading to optimization difficulties. Consequently, this can lead to sensitivity to hyperparameters and lower accuracy.

**Learning Difficulties of Different Input-Output Examples** We have observed that the learning difficulty varies among different output bits and input combinations. The training loss of different output bits shows different convergence speeds, indicating variations in difficulty. For inputs, the convergence speeds for different input samples on the same output bit exhibit substantial variations, challenging the assumption of independent and identically distributed (IID) samples. Detailed experimental results are in Appendix C.2.

## 5   A Regularized Triangle-Shaped Circuit Network Generation Framework

To address the aforementioned challenges, we have developed a novel Regularized Triangle-Shaped Circuit Network Generation Framework, namely T-Net. Our method comprises three modules: a multi-label transformation of training data, a triangle-shaped network architecture, and regularized training loss for efficient search and exact generation. Moreover, we propose an evolutionary algorithm assisted by a reinforcement learning agent restarting technique for efficient and effective neural circuit optimization. We defer more implementation details to Appendix D.

## 5.1 Multi-Label Transformation of Training Data

To address the scalability challenge posed by the exponential growth of truth tables with increasing input bit widths, we propose a novel approach: the multi-label transformation of training data, leveraging the Shannon decomposition theorem [37]. The Shannon decomposition theorem states that any boolean function (truth table) can be decomposed into two sub-functions (sub-tables) by selecting a decomposing variable. Formally, the theorem is expressed as:

$$f(X_1, \ldots, X_n) = X_i \cdot f_{1|X_i=1} + (!X_i) \cdot f_{2|X_i=0}, \tag{1}$$

where $f$ denotes the original boolean function, $X_i$ denotes the selected variable, $f_{1|X_i=1}$ and $f_{2|X_i=0}$ denote the decomposed sub-functions. Based on the key observation that the truth table exhibits a duality of Boolean functions, we first partition a large truth table into two smaller sub-tables by selecting a variable and fixing its value as 0 and 1, respectively.

After partitioning the truth table, a natural approach is to learn each sub-table separately. However, since the decomposed sub-tables share many logical nodes, individual learning prevents the active learning of these shared logical nodes. To overcome this challenge, we propose a multi-label data merge mechanism, which merges the two sub-tables into a multi-output table. This results in doubling the output number while halving the input number.

By recursively applying this partition-and-merge transformation, we can transform any large truth table into another truth table with significantly reduced input numbers and increased output numbers. Note that the input size significantly impacts the difficulty of neural circuit generation. Consequently, this transformation strategy provides two major advantages: 1) It enhances the scalability of our T-Net, enabling it to learn from truth tables with large input bit widths. 2) It significantly accelerates the learning process, as the learning difficulty of sub-tables is considerably reduced.

## 5.2 Triangle-Shaped Network Architecture

**Model Structure** Our network is structured as a neural network, where the neurons represent two-input NAND gates. During training, the neurons remain fixed while their connections are learned. An And-Inverter Graph (AIG) is a logic circuit composed of NAND gates and wires connecting the gates. By transforming the neurons and connections in the neural network into logic gates and wires, the neural network can be converted into an AIG circuit. Inspired by [5, 7], our basic differentiable circuit neural network structure is as follows. The network has depth $L$ and width $K$, indicating that it consists of $L$ layers, each comprising $K$ nodes. In this notation, $l = 0$ corresponds to the input of the circuit and $l = L + 1$ are the outputs. It is crucial to emphasize that the nodes in the output layer $(L + 1)$ are not considered gates; instead, they select the node within the network implementing the output signal. The network's inputs and outputs mirror the signals of a logical circuit, consisting of 0s and 1s. Our nodes has two inputs and one output as NAND gate. We denote the output of the $k^{th}$ neuron in the $l^{th}$ layer by $\mathbf{o}^{l,k}$. We denote the $p$-th input of the neuron (NAND gate) $\mathbf{o}^{l,k}$ by $\mathbf{i}_p^{l,k}$, where $p \in \{0, 1\}$. During the training phase, the discrete logic circuit undergoes a relaxation and continuousization process in two aspects. Firstly, the logical operations of logic gates are translated into their differentiable counterparts. For instance, $a \, NAND \, b$ is relaxed to $1 - (a \cdot b)$ [6]. Next, discrete network connections are parameterized, employing Gumbel-softmax [34] during forward propagation to continuousize and sample the connections between nodes, thus enabling optimization through gradient descent. Note that each neuron $o^{l,k}$ has two inputs $i_0^{l,k}$ and $i_1^{l,k}$, and can be connected to any neuron with layer number smaller than $l$ as its input neuron. We parameterize the connections of each neuron $o^{l,k}$ by a tensor of learnable parameters $\theta^{l,k} \in \mathbb{R}^{2 \times (l-1) \times K}$. Each parameter in the tensor $\theta_{p,i,j}^{l,k}$ represents the probability of connecting the $j^{th}$ neuron in the $i^{th}$ layer to the $p^{th}$ input of current neuron $o^{l,k}$. The computation of the $p^{th}$ input value for the neuron $o^{l,k}$ takes the form of

$$i_p^{l,k} := \sum_{i=0}^{l-1} \sum_{j=1}^{K} o^{i,j} \left[ \text{softmax} \left( \boldsymbol{\theta}^{l,k} \right) \right]_{p,i,j}, p = 0, 1 \tag{2}$$

$$o^{l,k} := 1 - \Pi_{p=0}^{1} i_p^{l,k} \tag{3}$$

During evaluation, each node's input selects only one connection based on the parameters, using maximization instead of softmax during forward calculation2, restoring discrete logic operations.

**Triangle-Shape** To fit the circuit bias on structure, we propose a Triangle-shaped network structure. Due to the inherent structural preference of logic circuits for a wider base and deeper top, the

commonly used rectangular network architecture is not well-suited for them. We employ a triangular structure that widens at the bottom layers to enhance expressive capability, thereby better fitting the foundational aspects of logic circuits. At the deep layers, the structure is narrower and deeper, which ensures adequate expressive power while reducing redundant nodes. This streamlined search space simplifies optimization, making it more manageable and efficient. Importantly, the last layer's width doesn't limit output diversity since any node can serve as an output. Our experiments confirm accurate generation even for circuits with many outputs (see Appendix E.4).

### 5.3 Regularized Training Loss towards Efficient Search and Exact Generation

**Regularized Skip-Connection** Note that for each node in the T-Net, it maintains a learnable probability distribution over all nodes in the T-Net whose layer number is smaller than this node. As shown in Figure 1(b), this distribution often overfits to shallow-layer nodes, causing too many skip connections. To prevent this, a natural solution is to enforce connections only to nodes in the previous layer. However, this significantly limits the search space, reducing expressive power (as demonstrated in the DNAS with no skip-connection in Table 1). To address this challenge, we propose a weighted regularization on the learnable probability distribution to softly suppress the probability of connecting to distant nodes across layers, while promoting connections to closer nodes. This approach avoids overfitting to excessive skip connections. Due to limited space, we defer the specifics of the weighting implementation to Appendix D.2.

**Boolean Hardness-Aware Loss** To alleviate the problem of extreme imbalance between positive and negative samples in the later stages of training, we introduce a Boolean Hardness-Aware Loss inspired by [38]. By weighting loss differently for various samples, these components help maintain training speed in the later stages. We also employed a temperature coefficient decay mechanism to reduce the discrepancy between continuous computation during training and discrete testing.

### 5.4 Neural Circuit Optimization

In this section, we introduce a novel optimization framework combined with our generation methods for a comprehensive logic synthesis approach. To optimize circuits, we use circuit equivalent transformations called operators, whose order and parameters significantly affect results. The goal is to find an optimal operator sequence that minimizes the circuit's size. Our method is an evolutionary algorithm optimization framework assisted by reinforcement learning with a restart strategy. The framework and more details can be seen in Appendix D.5

**Reinforcement Learning for Operator Sequence Optimization** Inspired by [39], our environment consists of the circuit, the logic synthesis tool ABC[40], and nine logic optimization operators. The agent receives the circuit state from the environment and outputs the next action, which includes an operator and its parameters. This operator is then applied to the circuit, resulting in the next circuit state. Ultimately, the RL model learns the optimal sequence of operators for the circuit, which is then used to optimize the circuit and hand it over to the EA.

**RL Agent Restart Strategy.** After a period of training, the agent parameters may converge and performance may settle into local optima. To address this, we restart the network parameters after a certain training period. Specifically, we reinitialize the agent parameters and recommence training using the optimal circuit while retaining the agent's encoder parameters. This helps escape from local optima and allows continued exploration of the search space for superior solutions. Retaining the encoder parameters preserves learned experiences, guiding subsequent training iterations.

**Evolutionary Algorithm Optimization Framework.** To better escape local optima, our optimization approach incorporates an Evolutionary Algorithm (EA) framework. The initial population consists of diverse circuits generated by our T-Net. To ensure that the generated circuits closely match functionality constraints with truth tables, we implement a legalization step to ensure functional compliance, as detailed in Appendix D.6. Subsequently, the EA iteratively optimizes circuits by the RL model and maintains an elite circuit population. Finally, the optimal circuit is selected as the output. Compared with only picking one optimal solution when restarting, EA can increase the diversity of the circuit and expand the search scope. A detailed procedure is in the Appendix D.5.

## 6 Experiments

Our experiments have four main parts. 1) Evaluate our generation and optimization approach on four open-source circuit benchmarks. 2) The scalability of our generation method. 3) Perform carefully

Table 1: Generation accuracy results. Impr. is the percentage decrease in wrong bits.

| Benchmark | | | | Basic DNAS | | DNAS Skip | | Darts- | | T-Net (Ours) | | |
|---|---|---|---|---|---|---|---|---|---|---|---|---|
| Size | Circuit | PI | PO | Acc.(%)↑ | Wrongs↓ | Acc.(%)↑ | Wrongs↓ | Acc.(%)↑ | Wrongs↓ | Acc.(%)↑ | Wrongs↓ | Impr.(%)↑ |
| Small | Espresso3 | 5 | 28 | 99.21 | 7 | 99.77 | 2 | 91.74 | 74 | 100 | 0 | 100 |
| | Espresso4 | 16 | 1 | 70.99 | 19008 | 100 | 0 | 77.44 | 14784 | 100 | 0 | 100 |
| | Espresso7 | 8 | 63 | 97.83 | 349 | 90.30 | 1564 | 97.16 | 458 | 100 | 0 | 100 |
| | LogicNets1 | 12 | 3 | 96.78 | 395 | 92.80 | 885 | 96.23 | 463 | 100 | 0 | 100 |
| | Random1 | 10 | 1 | 64.06 | 368 | 98.73 | 13 | 64.64 | 362 | 100 | 0 | 100 |
| | Arithmetic2 | 8 | 7 | 81.47 | 332 | 99.22 | 14 | 75.61 | 437 | 100 | 0 | 100 |
| Large | Espresso8 | 14 | 8 | 84.13 | 20797 | 86.86 | 17223 | 97.66 | 3056 | 100 | 0 | 100 |
| | Espresso9 | 14 | 14 | 93.82 | 14167 | 81.36 | 42756 | 97.29 | 6196 | 99.99 | 12 | 99.97 |
| | LogicNets4 | 12 | 3 | 79.17 | 2560 | 87.79 | 1500 | 83.77 | 1994 | 99.99 | 1 | 99.93 |
| | LogicNets6 | 12 | 3 | 65.34 | 4259 | 83.26 | 2057 | 96.05 | 485 | 100 | 0 | 100 |
| | Average | | | 82.51 | 3974.33 | 91.99 | 3995.11 | 86.31 | 1938.00 | **99.99** | **1.89** | **99.91** |

Table 2: Generation size results. Init Node is generated by SOP or our T-Net and Opt Node is optimized by resyn2. Impr. represents the percentage decrease in nodes achieved by our approach.

| Benchmark | | | | SOP+resyn2 | | Ours+resyn2 | | | |
|---|---|---|---|---|---|---|---|---|---|
| Size | Circuit | PI | PO | Init Node ↓ | Opt Node ↓ | Init Node ↓ | Impr.(%)↑ | Opt Node ↓ | Impr.(%)↑ |
| Small | Espresso3 | 5 | 28 | 205 | 149 | 155 | 24.39 | 136 | 8.72 |
| | Espresso4 | 16 | 1 | 129 | 66 | 37 | 71.32 | 27 | 59.09 |
| | Espresso7 | 8 | 63 | 482 | 296 | 334 | 30.71 | 275 | 7.09 |
| | LogicNets1 | 12 | 3 | 194 | 160 | 160 | 17.53 | 140 | 12.50 |
| | Random1 | 10 | 1 | 168 | 116 | 117 | 30.36 | 105 | 9.48 |
| | Arithmetic2 | 8 | 7 | 316 | 268 | 254 | 19.62 | 236 | 11.94 |
| Large | Espresso8 | 14 | 8 | 1159 | 965 | 207 | 82.14 | 151 | 84.35 |
| | Espresso9 | 14 | 14 | 1224 | 989 | 544 | 55.56 | 450 | 54.50 |
| | LogicNets4 | 12 | 3 | 808 | 670 | 636 | 21.29 | 601 | 10.30 |
| | LogicNets6 | 12 | 3 | 966 | 796 | 374 | 61.28 | 350 | 56.03 |
| | Average | | | 459.39 | 366.39 | **262.78** | **33.42** | **230.06** | **23.72** |

designed ablation studies to provide further insight into the DNAS-based circuit generation approach. 4) Verify the robustness of our approach through a sensitivity analysis.

**Benchmarks** We evaluate our approach using circuits from four benchmarks: Espresso[41], LogicNets[42], Random, and Arithmetic. Random circuits are random and decomposable Boolean functions, while Arithmetic circuits involve arithmetic functions with permuted inputs and dropped outputs. We selected 18 circuits (8 circuits are in Appendix E and the average are calculated by all 18 circuits), with inputs ranging from 7 to 16 and outputs from 1 to 63. Circuit sizes, based on node count synthesized through the SOP method, range from 100 to 1200. The circuits are divided into small (12 circuits, node count < 500) and large (6 circuits, node count > 500) datasets, highlighting synthesis challenges, especially for the large circuits.

**Experimental Setup** For circuit generation, we implemented our T-Net as in Section 5. We train our model on all input-output pairs of each circuit and evaluate their Boolean correctness. For circuit optimization, we use the RL model inspired by [39] and our EA framework. We conduct the LS operator sequence by open-source logic synthesis tool ABC[40]. Implementation details, hyperparameters, and hardware specifications can be found in Appendix E.

**Baselines** We compare our T-Net with the following generation approaches: 1) Basic DNAS: Based on [6], it learns connections but lacks skip-layer connectivity. 2) DNAS Skip: Proposed by Belcak et al.[5] and used by Google DeepMind in IWLS 2023[22]. We re-implemented this method with Gumbel-Softmax as Google did not open-source their code. 3) Darts-: An improvement on DNAS that addresses skip-connection issues in traditional NAS task[9]. We adapted it for circuit neural networks. 4) SOP (Sum-of-Products): A traditional LS method. We used resyn2[43] to optimize circuits synthesized by SOP and our T-Net, showing our method's superior initial solutions. On the other hand, the baselines for optimization include: 1) SOP with resyn2: Traditional LS method with the resyn2 operator. 2) IWLS Competition Results: We compare with the top three teams from 2022[31] and 2023[27]. These teams mostly used extensive traditional methods, while EPFL employing Bayesian optimization[30] and Google using the DNAS circuit generation method[7].

**Evaluation Metrics** We evaluate the accuracy and the size of the generated circuits and optimized circuits. 1) Accuracy: The ratio of correctly predicted output bits to the total number of output bits. Importantly, achieving 100% accuracy in the generated logic circuit stands as a fundamental criterion in the task of logic circuit synthesis. 2) Wrong Bits: The number of incorrectly predicted output bits, used to highlight accuracy differences in large-scale circuits. 3) Circuit Node/Size: The number of nodes in the generated AIG circuit, with fewer nodes being better for minimizing chip area.

Table 3: Optimization results. The term 'Impr.' is defined as the percentage decrease in the number of nodes achieved by our approach, relative to the traditional configuration.

| Benchmark | | | | Tradition | 2022 Teams | | | 2023 Teams | | | | Ours | |
|---|---|---|---|---|---|---|---|---|---|---|---|---|---|
| Size | Circuit | PI | PO | SOP+resyn2 | TUW | UCB | EPFL(AI) | NBU | EPFL(AI) | TUW | Google(AI) | Opt Node ↓ | Impr.(%)↑ |
| Small | Espresso3 | 5 | 28 | 149 | 77 | 88 | 79 | 83 | 81 | 70 | 72 | 69 | 53.69 |
| | Espresso4 | 16 | 1 | 66 | 25 | 25 | 25 | 25 | 25 | 25 | 25 | 25 | 62.12 |
| | Espresso7 | 8 | 63 | 296 | 147 | 151 | 152 | 146 | 148 | 138 | 141 | 139 | 53.04 |
| | LogicNets1 | 12 | 3 | 160 | 69 | 70 | 72 | 65 | 62 | 60 | 61 | 55 | 65.63 |
| | Random1 | 10 | 1 | 116 | 39 | 62 | 44 | 41 | 41 | 40 | 38 | 37 | 68.10 |
| | Arithmetic2 | 8 | 7 | 268 | 156 | 164 | 170 | 152 | 149 | 128 | 115 | 105 | 60.82 |
| Large | Espresso8 | 14 | 8 | 965 | 68 | 69 | 68 | 68 | 68 | 68 | 68 | 68 | 92.95 |
| | Espresso9 | 14 | 14 | 989 | 202 | 208 | 220 | 220 | 216 | 191 | 181 | 153 | 84.53 |
| | LogicNets4 | 12 | 3 | 670 | 340 | 240 | 246 | 285 | 241 | 206 | 167 | 154 | 77.01 |
| | LogicNets6 | 12 | 3 | 796 | 390 | 281 | 208 | 152 | 166 | 106 | 89 | 90 | 88.69 |
| | Average | | | 380.29 | 132.78 | 118.33 | 112.28 | 124.11 | 113.89 | 94.39 | 88.06 | **83.33** | **68.70** |

## 6.1 Comparative Evaluation

**Generation Evaluation** We evaluate generation accuracy and wrong bits across four datasets. The results in Table 1 show that T-Net significantly outperforms all baselines, achieving 100% accuracy on most circuits and at least 99.9% on all. T-Net shows an average accuracy improvement of 17.48% over Basic DNAS and 8% over DNAS Skip. Our approach for circuit neural networks also significantly outperforms general DNAS improvement methods like Darts-. Regarding wrong bits, even with similar accuracy, T-Net shows significantly fewer errors. For instance, in Espresso9, our method reduces wrong bits by 42,000. These results demonstrate the superiority of our approach.

In addition, we evaluate the generation circuit size and the optimized size after applying traditional operators. The results in Table 2 show that our method significantly outperforms the SOP method in initial node count, with an average improvement of 33.42%. Furthermore, circuits with fewer initial nodes also exhibit better optimization outcomes, with our method showing an average improvement of 23.72% in optimized nodes compared to traditional methods. Notably, for circuits with initial node improvements over 82%, we achieve up to 84.35% improvement in optimized nodes. These results highlight the effectiveness and superiority of our approach.

**Optimization Evaluation** We evaluate our proposed optimization framework, as shown in Table 3. By integrating our circuit generation method, we significantly reduced circuit size by 68.70% compared to the traditional SOP+resyn2 method, demonstrating our optimization approach's effectiveness. Additionally, our average size outperformed the 2022 first-place team, EPFL, by 25.78%, and the 2023 first-place team, Google, by 5.36%, significantly ahead of other teams. This highlights the superiority of our generation method and optimization framework in better circuit synthesis.

## 6.2 Scalability

To validate our multi-label transformation's effectiveness in improving accuracy, efficiency, and reducing circuit node count, we tested large circuits from the LogicNets and Espresso datasets. We use SOP to quickly test the truth table transformation method with different bits and select the bits with the smallest size. For LogicNets6, we decomposed it into four parts using two inputs. For Espresso9, we split the truth table into two parts.

Table 4: Scalability results of transformed truth table on large circuits.

| Circuit | Method | Nodes↓ | Time(h)↓ | Acc(%)↑ | Wrs.↓ |
|---|---|---|---|---|---|
| LogicNets6 | Default | 613 | 29 | 99.60 | 49 |
| | Decomp. | 374 | 14 | 100 | 0 |
| | Impv. | **39%** | **41%** | **0.4** | **100%** |
| Espresso9 | Default | 699 | 26 | 99.97 | 61 |
| | Decomp. | 610 | 16 | 99.99 | 6 |
| | Impv. | **13%** | **38%** | **0.02** | **90%** |

The experimental results in Table 4 show a 90% reduction in wrong bits and an average 26% decrease in nodes after transformation. For LogicNets6, the method reduced generation time by 41%. These results confirm the efficiency of our decomposition method for large circuits. By isolating complex variables, we mitigate circuit generation complexity, enhancing scalability. Despite doubling the number of output bits, our T-Net maintains high accuracy, indicating its robustness to increased output bits.

## 6.3 Ablation Study

In this section, we conducted an ablation study on four circuits to understand the contributions of each T-Net component. Our T-Net adds three modules to DNAS: regularized skip-connection, boolean hardness-aware loss, and T-Net, abbreviated in Table 5 as con., loss, and T, which leads to three main

Table 5: Ablation results on four benchmarks of circuit.

| Method | Small: Espresso5 (8 PI/4 PO) | | | | Large: LogicNets2 (12 PI/3 PO) | | | | Small: Arithmetic2 (8 PI/7 PO) | | | | Large: Random1 (10 PI/1 PO) | | | |
|---|---|---|---|---|---|---|---|---|---|---|---|---|---|---|---|---|
| | Acc.(%)↑ | Wrongs↓ | Nodes | Lev | Acc.(%)↑ | Wrongs↓ | Nodes | Lev | Acc.(%)↑ | Wrongs↓ | Nodes | Lev | Acc.(%)↑ | Wrongs↓ | Nodes | Lev |
| SOP | NA | NA | 172 | 13 | NA | NA | 292 | 17 | NA | NA | 316 | 12 | NA | NA | 168 | 19 |
| baseline | 93.94 | 277 | 145 | 17 | 79.40 | 2526 | 2 | 1 | 94.97 | 90 | 193 | 13 | 98.73 | 13 | 141 | 16 |
| +con. | 98.24 | 18 | 184 | 19 | 99.91 | 10 | 241 | 17 | 99.33 | 12 | 276 | 14 | 99.40 | 6 | 127 | 20 |
| +con.+loss | 100 | 0 | 172 | 18 | 100 | 0 | 281 | 21 | 100 | 0 | 263 | 15 | 100 | 0 | 155 | 20 |
| +con.+loss+T | 100 | 0 | 112 | 14 | 100 | 0 | 273 | 18 | 100 | 0 | 254 | 15 | 100 | 0 | 114 | 20 |

conclusions. 1) Compared to the baseline, the *con.* significantly enhances accuracy, highlighting the effectiveness of the regularized skip-connection module in mitigating skip-layer degradation and improving network expressiveness. 2) The *con.+loss* approach nearly doubles the reduction in wrong bits compared to +con., showing that the boolean hardness-aware loss function significantly boosts accuracy for challenging instances. 3) *con.+loss+T* improves both accuracy and node count, indicating that T-Net reduces training difficulty and enhances circuit generation effectiveness.

## 6.4 Sensitivity Analysis

We validate the sensitivity of our method to hyperparameters from two perspectives: random initialization and the initial size of the network. Experiments show that our approach uniformly maintained 100% accuracy across various random initialization and network initial sizes, underscoring its robustness to these fluctuations. Please see Appendix E.3 for details.

# 7 Conclusion

We rethink existing DNAS methods and empirically show three fundamental challenges pertaining to existing methods: 1) DNAS tends to overfit to too many skip-connection; 2) DNAS suffers from the structure bias between the network and the circuit's inherent structure, leading to inefficient search; 3) imbalanced learning across different input-output examples. Based on these insightful observations, we propose a noval neural logic gate network search framework, which has a Triangle-shaped structure, regularized skip-connection, and boolean hardness aware loss function. Experiments on four circuit benchmarks demonstrate that our method can precisely generate circuits with large AIG sizes. Moreover, our generated circuits have a significant 68% improvement in area surpassing the performance of the traditional method.

## Acknowledgments

We would like to thank all the anonymous reviewers for their insightful comments. This work was supported in part by National Key R&D Program of China under contract 2022ZD0119801, National Nature Science Foundations of China grants U23A20388, and 62021001. This work was supported in part by Huawei as well.

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

# A Related Work

As chip complexity has grown exponentially with the development of semiconductor technology, using machine learning (ML) to assist the automated chip design workflow has been an active topic of significant interest in recent years [1, 12, 44, 45]. The chip design workflow consists of many stages [1], such as high-level synthesis [46–48], logic synthesis [49–51], placement [45, 52, 53], design space exploration [54–57], etc. Our paper specifically targets ML applications in logic synthesis (LS), a pivotal area within Electronic Design Automation (EDA). We categorize this domain into two key segments: ML embedded in logic synthesis and ML for end-to-end logic synthesis.

## A.1 Machine Learning Embedded in Logic Synthesis

Machine Learning Embedded in Logic Synthesis refers to incorporating ML into a specific stage of LS, aiming to enhance the efficiency of LS and achieve improved Quality of Results (QoR). For example, [15] use machine learning to tune the optimization flow of LS operators; [14] use machine learning to predict key metrics after physical design and leverage the prediction to guide LS optimization; [18] use machine learning to improve decision-making in traditional LS methods. [17] apply machine learning to commonly used LS operators and significantly improve the efficiency of these operators. These embedded machine learning methods have significantly contributed to the advancement of LS.

## A.2 Machine Learning for End-to-end Logic Synthesis

Logic circuit synthesis is the process of producing a logic circuit that satisfies a given specification and is a classical problem in computer science [5]. In contrast to embedded methods that utilize machine learning to optimize specific stages within Logic Synthesis (LS), recent works propose employing machine learning to replace traditional logic synthesis stages, thereby achieving end-to-end logic synthesis. In the realm of logic circuit generation, scholarly efforts diverge into two primary approaches. Works like those of [20] and [21] explore this field through the lens of the languages employed to describe logic circuits. In contrast, other studies concentrate on the generation of logic circuits by conducting searches focused on the circuit structure itself. Notably, [6] integrates real-valued logic with continuous parameterization to create a differentiable logic gate network, enabling efficient training and rapid inference, crucial for end-to-end synthesis. [5] proposes a novel approach to directly search exact logic circuits by leveraging differentiable neural architecture search methods, offering significant potential in logic circuit synthesis. However, existing end-to-end methods show limitations in scaling to large circuits and sensitivity to hyperparameters. Based on these observations, we rethink the differentiable neural architecture search and propose a novel differentiable neural logic gate network search framework. An appealing feature of our method is that it is functionally complete and has low redundancy and high expressiveness.

## A.3 IWLS Contest

International Workshop on Logic & Synthesis (IWLS) is a global conference on logic synthesis that hosts a contest annually[8, 22]. The themes for the contests in 2022 [31] and 2023[27] were circuit synthesis from truth tables, scored based on the number of nodes, with fewer being better. This is the same setting our paper focuses on. The approaches of the participating teams are mostly based on traditional logic synthesis methods, involving both generation and optimization steps. The generation phase encompasses basic operators such as SOP [10, 11] and others, BDD-based methods[23–25], as well as methods utilizing mutual information [26], among others. Meanwhile, in 2023, Google DeepMind adopted a deep learning-based generation approach[7, 27]. They introduced a DNAS-based method to generate circuits, which, after three weeks of subsequent optimization, achieved first place in 2023. Since Google has not formally published their approach, we replicated it as a baseline for our study. We deeply analyzed and improved upon the DNAS-based generation method.

Optimizing circuits involves employing a series of circuit optimization operators. Given that the sequence of operator executions and their parameter configurations significantly impact the final quality of results, the primary task is to find the optimal operator sequence for a circuit. UCB team proposed new optimization scripts[28], while the TUW team introduced windowing methods to enhance the efficiency of SAT optimization methods[29]. The 2022 champion EPFL team utilized Bayesian optimization methods[30] with EA framework to optimize the operator sequence. Although both EPFL and we have employed the EA framework, what sets us apart is our utilization of RL methods with strong search capabilities and the introduction of a restart strategy to mitigate the impact of local optima. We achieved a 25% and 5% improvement over the results of the winners in the 2022 and 2023 competitions, respectively, through our comprehensive circuit synthesis approach.

## A.4 Reinforcement Learning (RL)

In general, standard RL fall into two categories: model-free RL [58–62] and model-based RL [63, 64]. In recent years, RL has achieved great success in many important real-world decision-making tasks [65, 66, 56, 67–69]. In this paper, we propose an evolutionary RL method for circuit optimization.

**Machine Learning for Combinatorial Optimization** Circuit generation and optimization is also essentially a combinatorial optimization problem. The use of machine learning to tackle combinatorial optimization problems has been an active topic of significant interest in recent years [70–76].

## B Background

### B.1 Logic Synthesis (LS)

The increasing complexity in modern chip design workflow demands innovations in Electronic Design Automation (EDA) to keep scalability without compromising Quality-of-Results (QoR). Multiple EDA tools are incorporated into the chip design workflow to synthesize, simulate, test, and verify different circuit designs efficiently and reliably [77]. Being at the top of most EDA tools, LS aims to transform a behavioral description of a design into an optimized gate-level circuit implementation. In general, LS consists of translation, circuit optimization, and technology mapping [78]. In the translation phase, a given boolean function, represented by a truth table, is transformed into a well-structured circuit. Then, in the circuit optimization phase, lots of logic optimization operators [79–81] are applied to an input circuit to optimize the circuit. Finally, in the technology mapping phase, the optimized logic circuit is mapped to an available technology library, e.g., a standard-cell netlist [82] or k-input lookup-tables [83]. In this paper, we propose a novel differentiable neural logic gate network search framework to synthesize logic circuits from input-output examples. This framework has the potential to replace the traditional phases of translation and circuit optimization, which introduces innovation to the realm of LS.

### B.2 Circuit Representation

Boolean Networks are well-studied discrete mathematical models with broad applications in chemistry, biology, circuit design, formal verification, etc. [33] A Boolean network is a directed acyclic graph (DAG), where nodes correspond to Boolean functions and directed edges correspond to wires connecting these functions. A Boolean function $f$ is a mapping from an n-dimensional space $B^n$ to a 1-dimensional space $B$, i.e. $f : B^n \rightarrow B$, where $B = \{0, 1\}$ denotes the Boolean domain. In the LS stage, a circuit is usually represented as And-Inverter Graphs (AIG), which is a concise and uniform representation of Boolean Networks. An AIG contains four types of nodes: the constant, primary inputs (PIs), primary outputs (POs), and two-input And (And2) nodes. Graph edges may be complemented, indicating a complemented signal. Given a node in AIG, its *fanins* are nodes connected by incoming edges of this node, and its *fanouts* are nodes connected by outgoing edges of this node. Each node has at most two incoming edges. The *PIs* are nodes without fanin, and the *POs* are nodes without fanout. The *size* of a circuit denotes the number of nodes in the AIG. The *depth (level)* of a circuit denotes the maximal length of a path from a PI to a PO in the AIG. The size and depth of a circuit are proxy metrics for the area and delay of the circuit, respectively.

### B.3 DNAS for Logic Synthesis

Differential Neural Network Architecture Search (DNAS) can be effectively applied for logic synthesis, where logic modules like AND gates are used as nodes, and connections between these nodes are optimized via gradient descent. The network's inputs and outputs mirror the signals of a logical circuit, consisting of 0s and 1s. During the training phase, the discrete logic circuit undergoes a relaxation and continuousization process in two aspects. First, the logical operations of logic gates are translated into their differentiable counterparts. For instance, $a\,AND\,b$ is relaxed to $a \cdot b$, and *NOT* $a$ is relaxed to $1 - a$ [6]. Next, discrete network connections are parameterized, employing Gumbel-softmax [34] during forward propagation to continuousize and sample the connections between nodes, thus enabling optimization through gradient descent to find high-quality solutions. In the evaluation stage, for each input of every logic module, only a single connection is chosen based on the parameters, and discrete logic operations are reinstated.

The DNAS method, as analyzed in Section 4, utilizes AND-NOT gates as nodes. The network's structure is defined by its depth $L$ and width $K$, signifying $L$ layers with $K$ nodes each. Here, $L = 0$ represents the circuit's input. It's important to note that the nodes in the output layer $(L + 1)$ are not gates and are excluded from the network count. They possess a singular input, selecting which

node in the network represents the output signal. Let $g^{l,k}$ denote the $k^{th}$ node in the $l^{th}$ layer, where $1 \leq l \leq L$ and $1 \leq k \leq K$. Let $in_p^{l,k}$ be the $p^{th}$ input of the node $g^{l,k}$, and $out^{l,k}$ be its output. The method to compute the value of the $p^{th}$ input for $g^{l,k}$ during training is as follows:

$$in_p^{l,k} := \sum_{i=0}^{l-1} \sum_{j=1}^{K} out^{i,j} \, \text{softmax} \left( c^{k,m,p} \right)_{i,j} \tag{4}$$

There are two common modeling forms for the connections: direct and direct-or-negation. Since the AND-NOT gate is sufficient to express all logical functions, learning the connections alone can fully represent the functionality of a circuit. However, this may require specific combinations of connections, which can be more easily accomplished using the NOT operation. Therefore, another modeling method allows edges to choose between negation or direct connection. In the motivation section of our paper, we utilize the direct connection as referenced in [5], while in our T-Net, we adopt the direct-or-negation connection.

## C  Motivation

### C.1  Settings

**Width and Depth of Networks.** In DNAS, the method for setting the initial size of the network is as follows. Assuming that a circuit synthesized by SOP has $N$ nodes and $L$ levels, the network width is set to $5 \times \lceil N/(5L) \rceil$, which is the average number of nodes per layer rounded up to the nearest multiple of five. The network depth is set to $5 \times \lfloor 2L/5 \rfloor$, which is twice the value of L, rounded down to the nearest multiple of five. In our T-Net, we set the total depth as DNAS, while the bottom block depth is 5 or 10 layers. We set the total nodes as same as the DNAS, and the upper block width smaller than the bottom. The network size of the two circuits in Section 4 are shown in Table 6.

Table 6: Network Size of the two circuits evaluated in motivation.

| Circuit | SOP | | DNAS | | T-Net | | | |
|---|---|---|---|---|---|---|---|---|
| | Node | Lev | Width | Depth | Width up | Width down | Depth up | Depth down |
| Small: Espresso7 (8 PI/63 PO) | 482 | 11 | 45 | 20 | 15 | 135 | 15 | 5 |
| Small: LogicNets1 (12 PI/3 PO) | 194 | 18 | 15 | 35 | 10 | 25 | 25 | 10 |

**Training setting** Both the training and validation datasets use the complete set of input-output combinations, meaning that if the input bit size is $K$, there are a total of $2^K$ input combinations. The batch size for the network is uniformly set to $2^{10}$. The number of training iterations is 100 thousand, and we report the optimal results evaluated during the training process.

### C.2  Motivation Experiment

**Hyperparameter Sensitivity Analysis** To verify the stability of generated circuits with different random initializations, we conduct tests using various random initializations. Specifically, we utilized four different seeds to evaluate two circuits from each of the datasets Espresso and LogicNets. Figure 5(a) illustrates the average accuracy and the fluctuation range of accuracy for these two circuits. The fluctuation range reached 14.5% accuracy, indicating that the accuracy of generated circuits is highly sensitive to random initialization, making it challenging to consistently produce stable results with DNAS.

**The Curse of Skip-Connection** We conduct a motivation experiment on the LogicNets circuit. As shown in Figure 3(b), the LogicNets circuit exhibits the same experimental phenomenon as the Espresso circuit in Section 4.2, where the output nodes are located in the shallow layers of the network, and skip connections bypass the majority of the nodes.

We also visualized the positions of circuit nodes of LogicNets within the network in Figure 3(c). Notably, this circuit has three outputs, resulting in layer configurations distinct from those in the single-output case shown in Figure 3(b). It can be observed that only a subset of bottom-layer nodes is integrated into the circuit, with over two-thirds of the nodes being left idle due to skip connections. This visual representation demonstrates that excessively distant skip connections diminish node utilization and the expressive capacity of the network.

Figure 4 illustrates the exploration degree of each network layer after convergence. A node is thus deemed 'explored' if it forms part of the discretized circuit at any stage during the training. The

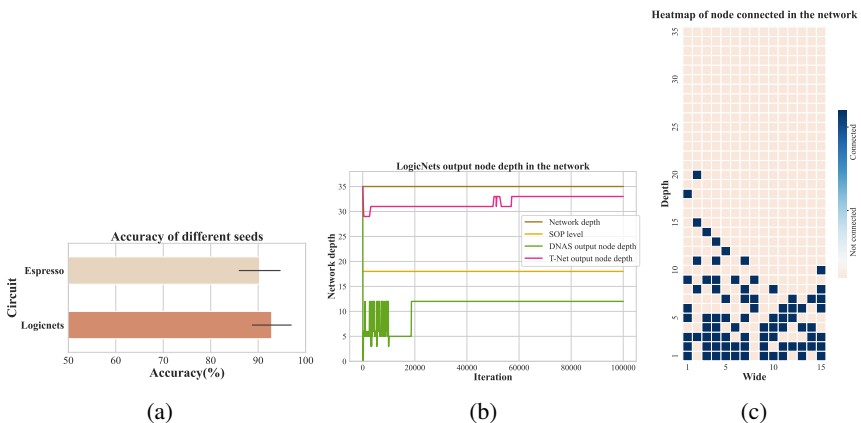

(a)                  (b)                  (c)

Figure 3: (a) Average accuracy and error bar of two circuits with four different initializations. (b) The depth for an output for another circuit. The curves labeled DNAS and T-Net in the graph represent the depth of the same output. Comparing network depth, it is evident that DNAS is only connected to very shallow layers, highlighting the curse of skip-connection. In contrast, our method learns deeper layers during subsequent training. (c) Network visualization of a Logicnet circuit after DNAS convergence. Dark nodes are circuit nodes.

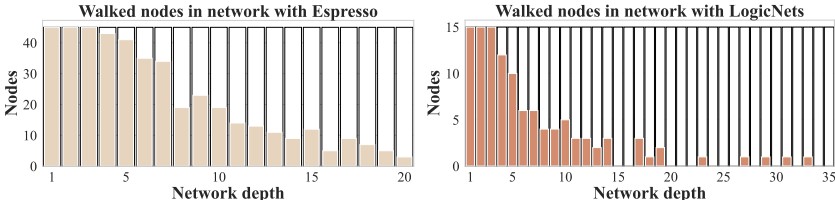

Figure 4: We present the exploration of two circuits. The height of the empty bars illustrates the total number of nodes in each layer, with the solid bars indicating the number of nodes that have been explored.

exploration degree is indicative of the number of nodes investigated within each layer. Observations reveal a higher exploration degree in the shallower layers, which progressively diminishes in the deeper layers. This pattern leads us to ponder whether a structural bias in circuit architecture contributes to the reduced exploration in the deeper layers of circuits.

**Learning Difficulties of Different Input-Output Examples** We have observed that the difficulty of learning varies among different output bits and input combinations.

To investigate variations in difficulty among different output bits, we analyzed training loss curves for various output bits within the same circuit. Using Mean Squared Error (MSE) loss, we visualized loss curves for three representative output bits from two circuits. From Figure 5(a) and (b), it's evident that the convergence speed of loss varies among different output bits—some converge quickly, while others exhibit slower descent. The distinct Boolean functions represented by different output bits naturally lead to varying levels of difficulty. Hence, during network training, addressing the challenges posed by harder-to-learn output bits is crucial.

Simultaneously, to investigate the learning difficulty among different *input* samples, we examined the convergence speeds for different inputs on the same output bit. Specifically, we recorded the iteration numbers at which the accuracy for each input sample reached 100 during a low iteration training process, representing the convergence time for each sample. Figure 5 (c) and (d) displays the distribution of convergence iterations for input-output examples involving three output bits. It can be observed that the convergence speed varies among different input samples. While the majority of samples converge within 10 thousand iterations, a small subset fails to converge correctly even after 100 thousand iterations. This observation underscores substantial variations in learning difficulty among various input-output pairings, challenging the conventional machine learning assumption of samples being independent and identically distributed (IID). This occurs because different input

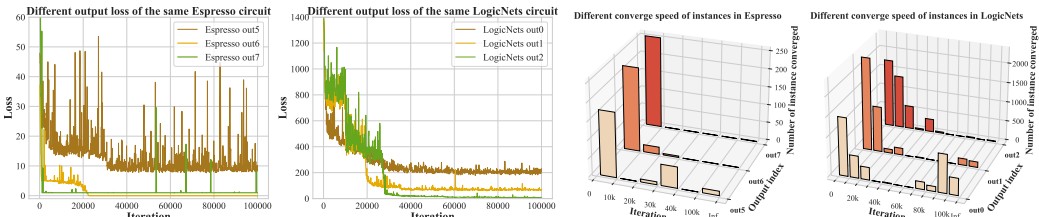

Figure 5: (a) and (b) Loss of different output in the same circuit. The convergence speed of loss varies among different outputs. (c) and (d) Distribution of convergence iterations for input-output examples. Different examples have different converge speeds. The bar between 10 thousand and Inf means these instances do not converge in 10 thousand iterations.

combinations of a circuit are interrelated, corresponding to parts of varying difficulty within the logical function.

## D    Implementation of Our Approach

### D.1    Implementation of T-Net

In our specific implementation, we adopted a simple two-block structure, consisting of a wide and shallow bottom rectangular block and a narrow and deep upper rectangular block. The size of the network is determined in relation to the scale of the circuit. The initial setting is based on a traditional square network. Assuming that a circuit synthesized by SOP has $N$ nodes and $L$ levels, the network width is set to $5 \times \lceil N/(5L) \rceil$, which is the average number of nodes per layer rounded up to the nearest multiple of five. The network depth is set to $5 \times \lfloor 2L/5 \rfloor$, which is twice the value of L, rounded down to the nearest multiple of five. In our T-Net, we maintain the total depth as in DNAS, with the bottom block having a depth of 5 or 10 layers. The total number of nodes is kept the same as in the traditional rectangular network, but the width of the upper block is smaller than that of the bottom.

### D.2    Regularized Skip-Connection

For the node $g^{l,k}$, its input computation with regulation is as follows:

$$in_p^{l,k} := \sum_{i=0}^{l-1} \left[ \sum_{j=1}^{K_i} out^{i,j} \, \mathrm{softmax}\left( unit^{l,k,p,i} \right)_j \right] \mathrm{softmax}\left( layer^{l,k,p} \cdot weight^l \right)_i \quad (5)$$

$weight^l$ having dimensions $1 \times l$, is non-learnable and solely varies with $l$. It functions similarly to a soft gating layer, with its values linearly increasing from $0.1$ to $1$ as per the formula:

$$\left( weight^l \right)_i = \begin{cases} 1 - 0.9\dfrac{i}{l-1} & 0 < i \leq 4 \\ 0.1 + 0.9\dfrac{i}{l-1} & 4 < i \leq l-1 \end{cases} \quad (6)$$

In the network's shallower layers, we configure the weight to preferentially form connections to the input layer. This design aims to augment the diversity of logical expression sub-formulas. Conversely, in the network's deeper layers, the weights are adjusted to favor connections with the more advanced nodes, thereby increasing the complexity of the logical expressions.

### D.3    Boolean Hardness-Aware Loss

Our hardness-aware loss has two parts, input-output loss and wrong-rate loss.

First, we use input-output loss to enhance the hard input-output examples from individuals. Specifically, we add a weight $w_{io}$ to each input-output loss.

$$w_{io} = e^{\alpha(|f(in)-out|-\delta)} \quad (7)$$

with $f(in)$ be the output by the net, $out$ be the label of output, $\alpha$ and $\delta$ be hyperparameter. The $|f(in) - out|$ represents the difference between the network output and the label, where a larger value indicates a more challenging sample. $\delta$ controls the boundary for challenging samples, while $\alpha$ regulates the extent of reinforcement .

Second, we use wrong-rate loss to enhance the hard input-output examples from global. $w_{wr} = \frac{\beta}{1-acc.}$ with $1 - acc$ being the wrong rate of the current network, which is updated every iteration. As $w_{wr}$ increases with the increase in accuracy, it dynamically maintains the loss at a similar magnitude, ensuring that the training speed does not decrease due to a low error rate. Then the loss function is

$$L(f(in), out) = w_{wr}\boldsymbol{w}_{io}\boldsymbol{L}_{MSE}(f(in), out) \tag{8}$$

### D.4 Temperature coefficient decay mechanism

We employ a decaying $\tau$ parameter. Due to the disparity between softmax calculations during training and the discretization during testing, there can be instances of accurate training but erroneous input-output combinations during testing. This phenomenon can be alleviated by reducing the hyperparameter $\tau$ in softmax. As $\tau$ increases, softmax approaches one-hot encoding, reducing the disparity between training and testing. However, a significant increase in $\tau$ may lead to highly discrete network parameters, making training extremely challenging. Therefore, it is crucial to set an appropriate size for $\tau$. We design a decay mechanism for $\tau$, continuously decreasing it as accuracy improves, ensuring a trade-off between training effectiveness and speed.

### D.5 Optimization

**Action Space** We use RL agent to search optimal operator sequence. As shown in Table 7, the action space has 9 operators and their corresponding 16 parameters. For each parameter of an operator, its range is denoted as [a, b] in the table, showing that the parameter can be any integer between a and b (inclusively).

**State Representation**. The state in the environment has three parts: a set of metrics that depict the current circuit design, the raw AIG graph, and the historical actions. The first part is a feature vector consisting of the following values at step t: Number of logic gates and logic depth. The current length of the operator sequence. For each step, we record the selected operator and parameters as a normalized vector and form the historical action vector.

Table 7: The action space in our environment, with parameter name '-x' and the range of the continuous parameter [a, b].

| Operator | Parameter |
|----------|-----------|
| &st | N/A |
| &blut | -m -r -a -C[1,8] |
| &b | -s -d |
| &dsdb | -C[8,400] |
| &sopb | -C[8,400] |
| &if -g | -C[8,400] |
| &dc2 | -l |
| &dch | -x -W[8,512] -C[1000,10000] -S[5000,50000] |
| &transtoch | -M [1:4] -R[1:200] |

**Evolutionary Algorithm Framework** Our Evolutionary Algorithm (EA) Framework involves several key steps as shown in Figure 6. Firstly, the exploration space can be defined as circuits of different structures with the same logical functionality. The initial population size is denoted as $P$, comprising diverse circuits generated by our T-Net. These $P$ circuits, serving as the initial solutions for the RL model, undergo a period of training. Following this training phase, each circuit yields $Q = M/P$ optimized circuits. Here, mutation is defined as circuit optimization. It is important to note that the optimized circuits resulting from different initial circuits are collectively input into the model without differentiation. Subsequently, the $M$ optimized circuits are sorted based on their node count, and the top-performing $P$ circuits are selected as the next generation. This iterative process continues, evolving the population until the optimal circuit is identified as the final result. Compared with only picking one optimal solution when restarting, EA can increase the diversity of the circuit and expand the search scope.

### D.6 Legalization

For circuits that are not generated with complete precision, we adopt a legalization approach to correct the few erroneous bits in the logic expression. Here, *bit* refers to a combination of a set of inputs and

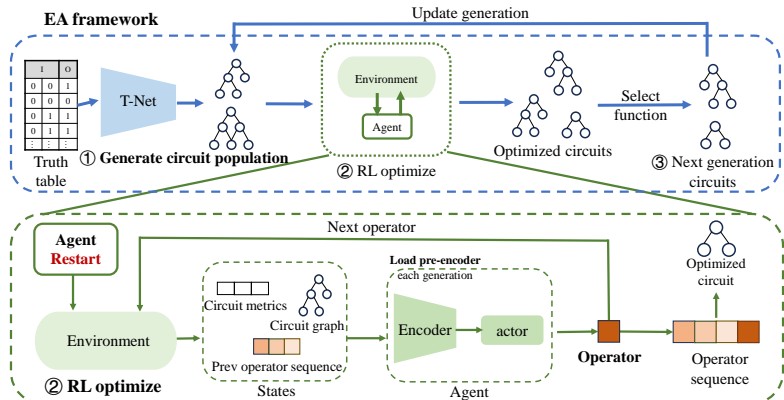

Figure 6: Framework of the evolutionary algorithm assisted by RL agent restarting technique for circuit optimization.

Table 8: Generation accuracy results. Impr. is the percentage decrease in wrong bits.

| | Benchmark | | | Basic DNAS | | DNAS Skip | | Darts- | | T-Net (Ours) | | |
|---|---|---|---|---|---|---|---|---|---|---|---|---|
| Size | Circuit | PI | PO | Acc.(%)↑ | Wrongs↓ | Acc.(%)↑ | Wrongs↓ | Acc.(%)↑ | Wrongs↓ | Acc.(%)↑ | Wrongs↓ | Impr.(%)↑ |
| Small | Espresso1 | 10 | 4 | 98.43 | 64 | 79.95 | 821 | 97.83 | 89 | 100 | 0 | 100 |
| | Espresso2 | 7 | 3 | 66.67 | 128 | 100 | 0 | 70.31 | 114 | 100 | 0 | 100 |
| | Espresso5 | 8 | 4 | 72.95 | 277 | 97.65 | 24 | 72.75 | 279 | 100 | 0 | 100 |
| | Espresso6 | 9 | 5 | 78.67 | 546 | 97.85 | 55 | 80.47 | 500 | 100 | 0 | 100 |
| | LogicNets2 | 12 | 3 | 85.73 | 1754 | 85.92 | 1730 | 84.91 | 1854 | 100 | 0 | 100 |
| | Arithmetic1 | 12 | 3 | 79.56 | 2880 | 98.73 | 156 | 88.61 | 1400 | 100 | 0 | 100 |
| Large | LogicNets3 | 12 | 3 | 90.17 | 1207 | 90.09 | 1218 | 88.53 | 1409 | 99.9 | 12 | 99.01 |
| | LogicNets5 | 12 | 3 | 80.14 | 2440 | 84.59 | 1894 | 92.54 | 930 | 99.93 | 9 | 99.52 |
| | Average | | | 82.51 | 3974.33 | 91.99 | 3995.11 | 86.31 | 1938.00 | **99.99** | **1.89** | **99.91** |

a single output. Specifically, we represent the input combinations using the method of minterms, and then integrate them into the output bit. For example, if the circuit has three inputs $a, b$ and $c$, and the output for the input combination *101* should be *1*, then the subtree $ab'c$ is combined with the original output node through an *OR* operation, forming a new output node. This legalization process can complete cases that are only a few bits short. However, this approach involves a significant amount of redundancy and is not suitable for situations where there are many errors.

# E  Experiment

## E.1  Setting

**Hyperparameter** The batch size is set to 1024, with the learning rate set at 0.02. The temperature coefficient $\tau$ starts at 1 and decays to 0.5 when the accuracy approaches 100%. The training process lasts for 100 thousand iterations, and the model with the highest evaluation score is selected as the final result. We use Sum Squared Errors instead of the common Mean Squared Errors(MSE) to ensure that the loss does not become too small in later stages. In Equation 7, the hyperparameter $\alpha$ is set to 2, and $\delta$ is set to 0.3. In $w_{wr}$, $\beta$ is set to 10.

The size of the circuit generation network varies according to the circuit. For DNAS, the network size is determined in relation to the scale of the circuit. Assuming that a circuit synthesized by SOP has $N$ nodes and $L$ levels, the network width is set to $5 \times \lceil N/(5L) \rceil$, which is the average number of nodes per layer rounded up to the nearest multiple of five. The network depth is set to $5 \times \lfloor 2L/5 \rfloor$, which is twice the value of L, rounded down to the nearest multiple of five. In our T-Net, we maintain the total depth as in DNAS, with the bottom block having a depth of 5 or 10 layers. The total number of nodes is kept the same as in the traditional rectangular network, but the width of the upper block is smaller than that of the bottom.

We train our method with ADAM [84] using the PyTorch.

**Hardware specification** Our experiments were conducted on a Linux-based system powered by a 3.60 GHz Intel Xeon Gold 6246R CPU and NVIDIA RTX 2080 GPU.

## E.2  Main Evaluation

The other 8 circuits mentioned in the main evaluation are shown in Table 8, 9 and 10.

Table 9: Generation size results. Init Node is generated by SOP or our T-Net and Opt Node is optimized by resyn2. Impr. represents the percentage decrease in nodes achieved by our approach.

| Benchmark | | | | SOP+resyn2 | | Ours+resyn2 | | | |
|---|---|---|---|---|---|---|---|---|---|
| Size | Circuit | PI | PO | Init Node ↓ | Opt Node ↓ | Init Node ↓ | Impr.(%)↑ | Opt Node ↓ | Impr.(%)↑ |
| Small | Espresso1 | 10 | 4 | 172 | 130 | 139 | 19.19 | 123 | 5.38 |
| | Espresso2 | 7 | 3 | 116 | 85 | 90 | 22.41 | 70 | 17.65 |
| | Espresso5 | 8 | 4 | 172 | 120 | 105 | 38.95 | 96 | 20.00 |
| | Espresso6 | 9 | 5 | 187 | 133 | 140 | 25.13 | 121 | 9.02 |
| | LogicNets2 | 12 | 3 | 292 | 257 | 254 | 13.01 | 219 | 14.79 |
| | Arithmetic1 | 12 | 3 | 139 | 107 | 128 | 7.91 | 116 | -8.41 |
| Large | LogicNets3 | 12 | 3 | 599 | 490 | 445 | 25.71 | 376 | 23.27 |
| | LogicNets5 | 12 | 3 | 941 | 798 | 611 | 35.07 | 549 | 31.20 |
| | Average | | | 459.39 | 366.39 | 262.78 | 33.42 | 230.06 | 23.72 |

Table 10: Optimization results. The default results are synthesized by the traditional SOP method. We use optimization operators to synthesize the circuits as the Opt Node shows. The term 'Impr.' is defined as the percentage decrease in the number of nodes achieved by our approach, relative to the default configuration.

| Benchmark | | | | Tradition | 2022 Teams | | | 2023 Teams | | | | Ours | |
|---|---|---|---|---|---|---|---|---|---|---|---|---|---|
| Size | Circuit | PI | PO | SOP+resyn2 | TUW | UCB | EPFL(AI) | NBU | EPFL(AI) | TUW | Google(AI) | Opt Node ↓ | Impr.(%)↑ |
| Small | Espresso1 | 10 | 4 | 130 | 58 | 64 | 59 | 67 | 56 | 52 | 51 | 47 | 63.85 |
| | Espresso2 | 7 | 3 | 85 | 28 | 35 | 28 | 28 | 28 | 28 | 28 | 28 | 67.06 |
| | Espresso5 | 8 | 4 | 120 | 37 | 47 | 37 | 37 | 37 | 37 | 37 | 37 | 69.17 |
| | Espresso6 | 9 | 5 | 133 | 46 | 58 | 48 | 46 | 46 | 46 | 44 | 46 | 65.41 |
| | LogicNets2 | 12 | 3 | 257 | 134 | 134 | 138 | 133 | 129 | 118 | 112 | 108 | 57.98 |
| | Arithmetic2 | 8 | 7 | 268 | 156 | 164 | 170 | 152 | 149 | 128 | 115 | 105 | 60.82 |
| Large | LogicNets3 | 12 | 3 | 490 | 153 | 144 | 142 | 157 | 140 | 138 | 128 | 123 | 74.90 |
| | LogicNets5 | 12 | 3 | 798 | 354 | 215 | 212 | 463 | 355 | 187 | 172 | 165 | 79.32 |
| | Average | | | 380.29 | 132.78 | 118.33 | 112.28 | 124.11 | 113.89 | 94.39 | 88.06 | 83.33 | 68.70 |

## E.3 Sensitivity Analysis

We validate the sensitivity of our method to hyperparameters from two perspectives: random initialization and the initial size of the network.

**Random initialization.** we executed repeated experiments with a diverse set of random initializations. This entailed testing a single circuit from each of the four circuit categories using three distinct random initializations. As Table 11 elucidates, our method uniformly maintained 100% accuracy across various random initializations, underscoring its robustness to these fluctuations. Furthermore, the disparity in circuit depth synthesized under different random initializations remained below 20%, showcasing our method's remarkable stability in network depth. This consistent performance helps prevent the descent into local optima, often characterized by an excessive number of cross-layer connections.

**Network initial size.** We evaluate one circuit with different network initial sizes. Specifically, we change the depth and width of the two network blocks up and down. Table 12 demonstrates that our method consistently generates accurate circuits across a range of network initial sizes, thereby evidencing the robustness of our approach to variations in network size.

Table 11: Sensitivity analysis on different initialization.

| Method | Small: Espresso5 (8 PI/4 PO) | | | Small: LogicNets2 (12 PI/3 PO) | | | Small: Arithmetic2 (8 PI/7 PO) | | | Large: Random1 (10 PI/1 PO) | | |
|---|---|---|---|---|---|---|---|---|---|---|---|---|
| | Acc(%)↑ | Nodes↓ | Lev | Acc.(%)↑ | Nodes | Lev | Acc(%)↑ | Nodes↓ | Lev | Acc(%)↑ | Nodes↓ | Lev |
| SOP | NA | 172 | 13 | NA | 292 | 17 | NA | 316 | 12 | NA | 168 | 19 |
| T-Net Init1 | 100 | 117 | 14 | 100 | 258 | 20 | 100 | 269 | 13 | 100 | 126 | 15 |
| T-Net Init2 | 100 | 176 | 18 | 100 | 267 | 20 | 100 | 258 | 13 | 100 | 150 | 18 |
| T-Net Init3 | 100 | 108 | 17 | 100 | 281 | 23 | 100 | 287 | 15 | 100 | 139 | 15 |

**Network Depth.** Deeper networks are not always better. We demonstrate in the following table the experimental results of a circuit at different network depths. As can be seen from Table 13, when the network is too shallow, it lacks sufficient expressive capacity, thus failing to generate exact circuits; conversely, when the network is too deep, it tends to produce relatively redundant circuits. Therefore, the depth of the network should be as shallow as possible while having enough expressive power to minimize the size of the generated circuit. In our experiments, we rounded down the depth of the SOP circuit to 5 as the network depth.

Table 12: Sensitivity analysis on different initial network sizes.

| Small: LogicNets2 (12 PI/3 PO) | | | | |
|---|---|---|---|---|
| Width up | Width down | Depth up | Depth down | Accuracy |
| 10 | 40 | 20 | 10 | **100** |
| | | 15 | 10 | **100** |
| | | 25 | 10 | **100** |
| | | 20 | 5 | **100** |
| | | 20 | 15 | **100** |
| 5 | 40 | 20 | 10 | **100** |
| 15 | 40 | | | **100** |
| 10 | 30 | | | **100** |
| 10 | 50 | | | **100** |

Table 13: Sensitivity analysis on different network depth.

| Small: LogicNets2 (12 PI/3 PO) | | | | |
|---|---|---|---|---|
| Method | Network Depth | Acc.(%) | Node | Level |
| Default (SOP) | N/A | N/A | 292 | 17 |
| T-Net | 15 | 99.05 | 184 | 13 |
| | 20 | 99.86 | 213 | 16 |
| | 25 | 100 | 253 | 17 |
| | 30 | 100 | 258 | 20 |
| | 35 | 100 | 321 | 20 |

## E.4 Other Experiments

**Triangle shape is adaptable to circuits with more POs.** The output layer of the network is a selection layer, allowing any node within the network to be chosen as an output. Consequently, the number of nodes in the final gate layer does not limit the choices for outputs. We conducted an experiment to demonstrate this. For two circuits with extreme multiple POs, we use T-Net and a rectangular network with the same base width as T-Net to generate circuits. As shown in Table 14, even when the width of the upper layers of the network is much smaller than the number of outputs, it is still able to generate circuits exactly with fewer nodes compared to a rectangular network.

Table 14: Analysis T-Net on circuits with extreme multiple POs.

| Circuit | Espresso3 (PI 5/ PO 28) | | | Espresso7 (PI 8/ PO 63) | | |
|---|---|---|---|---|---|---|
| | Network Width | Acc.(%) | Node | Network Width | Acc.(%) | Node |
| Rectangular Net | 50 | 100 | 206 | 135 | 100 | 411 |
| T-Net | 20 up/ 50 down | 100 | 174 | 15 up/ 135 down | 100 | 377 |

## F Limitation

Our approach for circuit generation and optimization relies on GPU for training, while the majority of existing logic synthesis tools operate in CPU environments. Consequently, our method may not be well-suited for deployment in CPU environments.

## G Broader Impact

**Academic Impact** This paper provides a thorough analysis of the application of differentiable neural architecture search (DNAS) methods to circuit generation. The insights may have positive implications for future work utilizing DNAS for circuit generation.

**Social Impact** This paper explores the optimization of logic circuits. By employing ML methods for circuit generation and optimization, it becomes feasible to enhance chip design quality and reduce costs in comparison to manual design processes.

## H Licence

We include the following licenses for the code, benchmarks we used in this paper.

**Benchmarks**: Espresso[41]: Copyright, LogicNets[42]: Licence, Random and Arithmatic[31]: Licence. **Code**: DNAS[6]:Licence.

