# OpenReview forum: "Towards Next-Generation Logic Synthesis: A Scalable Neural Circuit Generation Framework"
_NeurIPS.cc/2024/Conference — NeurIPS 2024 poster_

### Official Review · Reviewer_qf83 · 2024-07-12

**Soundness:** 3
**Presentation:** 2
**Contribution:** 3
**Rating:** 6
**Confidence:** 3

**Summary:**

The paper studies the application of differentiable neural architecture search (DNAS) to the problem of logic synthesis from input-output examples. The authors first analyze several challenges when directly applying existing DNAS methods to the problem. Based on those findings three modifications are proposed: (1) A transformation of the input-output examples that decreases the number of input and increases the number of output variabels. (2) Changing the network search space from rectangular to triangular shaped to be more aligned with typical network shapes. (3) Adding a regularization to avoid overfitting to shallow circuits and adding weights to the loss of positive and negative examples. In addition to adapting DNAS methods, the authors present a neural circuit optimization postprocess which learns a sequence of circuit operations to optimize the size of the circuit. The sequence is learned with reinforcement learning and an evolutionary algorithm. In an experimental evaluation, the authors demonstrate that both the modification of the DNAS method and the circuit optimization postprocess yield large gains in terms of correctness and circuit size.

**Strengths:**

- Recent International Workshop on Logic & Synthesis (IWLS) contest results sparked interest in the application of neural architecture search (NAS) to logic synthesis. To the best of my knowledge, this paper is the first to extensively study this promising combination. The paper first presents valuable insights into the challenges of applying the NAS methods as is. Derived from those challenges the authors propose well-motivated changes to the standard methods. In the experimental evaluation, it is shown that each change either contributes to improving the accuracy or decreasing the size of the resulting circuit.
- As part of the framework, the authors introduce a novel circuit optimization based on learning a sequence of circuit operations with reinforcement learning and an evolutionary algorithm. This optimization step does not only seem valuable in combination with the introduced T-Net but also as an independent postprocess on top of other logic synthesis techniques.
- The size of the circuits is evaluated in comparison with the top teams from recent IWLS contests. The winners of IWLS can be considered state-of-the-art methods and the experimental evaluation demonstrates substantial improvements over them. To perform this comparison the authors re-implemented some of the approaches which have not been open-sourced. If the authors were to make their code publicly available this would also be a valuable contribution to the research community.

**Weaknesses:**

- Large parts of the paper rely on previous work for the background of the problem and the method itself. For example, the paper does not formally introduce the logic synthesis problem from input-output examples, nor does the paper introduce the notion of differentiable neural architecture search. The exact method of relaxing a logic neural network for training and discretizing for evaluation only becomes evident in the second half of the paper (Section 5.2), making it difficult to understand the first half on motivating challenges. Another difficulty in following the paper is that the authors refer to the appendix for many aspects of the approach. For example, experiments on the motivating challenges, the implementation of the newly introduced regularization and weighted loss, and details on the circuit optimization in general can only be found in the appendix.
- Since the authors already compare with the top winners from the IWLS contests, it is not clear to me why they do not evaluate the IWLS benchmarks. Instead, the authors choose a smaller (potentially less challenging) set of benchmarks without providing insights into how the benchmarks were selected.

**Questions:**

Is the circuit optimization evaluated on top of T-Net or is the input circuit obtained in a different way?

**Limitations:**

The authors only discuss the GPU requirement as a limitation such that important aspects are missing from the discussion. For example, even though T-Net largely improves on accuracy compared to previous methods, it is not guaranteed to reach perfect accuracy.

---

> ### Author Rebuttal · Authors · 2024-08-07
>
> # Response to Reviewer qf83
> We thank the reviewer for the insightful and valuable comments. We respond to each comment as follows and sincerely hope that our rebuttal could properly address your concerns. If so, we would deeply appreciate it if you could raise your score. If not, please let us know your further concerns, and we will continue actively responding to your comments and improving our submission.
>
> ## Weakness 1.
>
> > **1. The paper does not formally introduce the logic synthesis problem from input-output examples and the notion of differentiable neural architecture search.**
>
> Thanks for the valuable suggestions. We have revised the Background Section as follows.
>
> **Formulation of LS from IO examples** In recent years, synthesizing circuits from IO examples has gained increasing attention [1][2][3]. Specifically, researchers aim to use machine learning to generate a circuit based on a truth table that describes the circuit's functionality. Each line in the truth table represents an input-output pair, indicating the output produced by the circuit for a given input. In the machine learning domain, researchers formulate the truth table as a training dataset comprising many input-output pairs and use an ML model to generate circuits that accurately fit the dataset.
>
> **DNAS for LS from IO Examples** Recent works [1][2] propose leveraging traditional DNAS methods for generating circuit graphs from IO examples, showing a promising direction for next-generation logic synthesis. Specifically, they formulate a neural network as a circuit graph, where each neuron represents a logic gate and connections between neurons represent wires connecting these logic gates. For a parameterized neural network, the neurons are fixed as logic gates, and the connections between neurons are parameterized as learnable parameters. To enable differentiable training via gradient descent, continuous relaxation is introduced into discrete components of the neural network. First, the logical operations of logic gates (neurons) are translated into their differentiable counterparts. For example, $a \ \textit{AND} \ b$ is relaxed to $a\cdot b$ [4]. Second, discrete network connections are parameterized using Gumbel-softmax [5].
>
> > **2. Another difficulty in following the paper is that the authors refer to the appendix for many aspects of the approach.**
>
> Thanks for the valuable suggestion. We will revise our manuscript by retaining the key content in the main text and moving the minor content to the appendix. Specifically, we will update the "Background," "Motivation," and "Method" Sections as follows.
>
> For Background, we present the problem formulation of **logic synthesis (LS) from input-output examples**, and details of **the traditional DNAS approach for LS**.
>
> For Motivation, we first present the main challenge of the traditional DNAS for LS that the method **struggles to generate circuits accurately**, especially for large circuits. We then present two major reasons for this challenge: **the curse of skip-connection** and **the structure bias of circuits**.
>
> For Method, we first present the three key modules for neural circuit generation. 1) To reduce the learning complexity of **large circuits**, we present the multi-label transformation module to decompose a large truth-table (dataset) into several small sub-truth-tables (sub-datasets). 2) To address **the curse of skip-connection challenge**, we present details of the regularized skip-connections module. 3) To **leverage the structure bias of circuits**, we present details of the triangle-shaped network architecture. We then present details of our circuit optimization approach and **provide the pseudocode of the algorithm**.
>
> ## Weakness 2.
>
> > **3. Since the authors already compare with the top winners from the IWLS contests, it is not clear to me why they do not evaluate the IWLS benchmarks.**
>
> Thanks for the valuable suggestion. We have compared our method with Google DeepMind's contest results on **30 more circuits** from the IWLS benchmark as shown in Table 2 in the attached pdf in Global Response. The results show that our method **achieves 3.03% node reduction** compared with Google DeepMind's contest results.
>
> In addition, the 18 circuits used in the main text are indeed sourced from the IWLS benchmark as well, and we compared our method with Google DeepMind's contest results on these circuits in the main text (achieving 5.36% node reduction).
>
> ## Question 1:
>
> > **4. Is the circuit optimization evaluated on top of T-Net or is the input circuit obtained in a different way?**
>
> Yes, the results of our method are obtained by optimizing circuits generated using T-Net.
>
> ## Limitation 1.
>
> > **5. T-Net is not guaranteed to reach perfect accuracy.**
>
> To ensure perfect accuracy in our circuit generation, we applied a legalization method to our final generated circuits as detailed in Appendix D.6.
>
> ## Open-source code.
>
> > **6. If the authors were to make their code publicly available this would also be a valuable contribution to the research community.**
>
> Yes, we will make our code publicly available once our paper is accepted.
>
> [1] Designing better computer chips. Google DeepMind, 2023, https://deepmind.google/impact/optimizing-computer-systems-with-more-generalized-ai-tools/.
>
> [2] Peter Belcak, et al. Neural combinatorial logic circuit synthesis from input-output examples. NeurIPS Workshop, 2022.
>
> [3] IWLS Programming Contest Series Machine Learning + Logic Synthesis. IWLS, 2024, https://www.iwls.org/contest/.
>
> [4] Petersen, et al. Deep differentiable logic gate networks. NeurIPS, 2018.
>
> [5] Jang, et al. Categorical Reparametrization with Gumble-Softmax. ICLR, 2017.

---

> ### Author Response · Authors · 2024-08-12
> **Response to Reviewer qf83--Looking forward to your further feedback**
>
> Dear Reviewer qf83,
>
> We are writing as the authors of the paper "Towards Next-Generation Logic Synthesis: A Scalable Neural Circuit Generation Framework" (ID: 17130).
>
> We sincerely thank you once more for your insightful comments and kind support! We are writing to gently remind you that **the deadline for the author-reviewer discussion period is approaching** (due on Aug 13). We eagerly await your feedback to understand if our responses have adequately addressed all your concerns. *If so, we would deeply appreciate it if you could raise your score*. If not, we are eager to address any additional queries you might have, which will enable us to enhance our work further.
>
> Once again, thank you for your guidance and support.
>
> Best,
>
> Authors

---

> ### Comment · Reviewer_qf83 · 2024-08-12
>
> I would like to thank the authors for their text revisions and the evaluation of the additional IWLS benchmarks. I believe the proposed revisions will improve the presentation of the paper. The evaluation of the additional benchmark further strengthens the experimental results. However, the additional benchmarks are again a subset of the IWLS competition benchmarks. I am still concerned that the method is not compared on the full benchmark set.

---

> > ### Author Response · Authors · 2024-08-12
> > **Evaluation on the full IWLS benchmark set (1/3)**
> >
> > Dear Reviewer qf83
> >
> > We would like to extend our sincere gratitude for the time and effort you have devoted to reviewing our submission. Your insightful comments and constructive suggestions have been invaluable to us, guiding us in improving the quality of our work!
> >
> > > **Remark**: Since the rebuttal phase, we **have been actively expanding our experiments to include the entire IWLS benchmark**. However, due to **limited time and computational resources**, we have to manage to keep the optimization time for each circuit **within a week**. In contrast, the SOTA method from Google, which we are comparing against, is reported to take **3 weeks per circuit** to optimize.
> >
> > We have presented the results of our method on the **full benchmark** compared to the top IWLS winners in Tables 1 and 2 as follows. As shown in Table 1, our method **significantly outperforms the top IWLS winners**, including the IWLS 2022 first-place team (EPFL), the IWLS 2023 first-place team (Google), and the IWLS 2023 second-place team (TUW), **in terms of the number of Wins** (smaller circuit sizes). Moreover, Table 2 provides a detailed comparison for each circuit, demonstrating that our method **reduces circuit sizes** by an average of **8.78%** compared to SOP, **17.02%** compared to the IWLS 2022 first-place team (EPFL), and **10.7%** compared to the IWLS 2023 second-place team (TUW).
> >
> > Our method does not fully outperform the IWLS 2023 first-place team (Google) due to **limited optimization time** (**1 week versus 3 weeks**). Nevertheless, it's important to note that when excluding just five corner cases, our method **achieves comparable circuit sizes to the IWLS 2023 first-place team while using only one-third of the optimization time**, highlighting the strong performance of our approach.
> >
> > We sincerely hope that our results on the full IWLS benchmark has adequately addressed your concerns. **If so, we would deeply appreciate it if you could raise your score**. If there are any further questions or concerns, we would be more than willing to address them in order to further enhance the quality of our submission.
> >
> > Table 1. We report **the number of Wins** of our method (smaller circuit sizes) compared to the IWLS 2022 first-place team (EPFL), the IWLS 2023 first-place (Google) and second-place (TUW) teams, on the **full IWLS benchmark** set.
> > |  | Generation | Optimization |  |  |
> > |:---:|:---:|:---:|:---:|:---:|
> > |  | SOP | EPFL | TUW | Google |
> > | **Ours Wins** | **75**/100 | **80**/100 | **60**/100 | **43**/100 |
> > | **Ours Ties** | 3/100 | 17/100 | 27/100 | 29/100 |
> > | **Ours Loses** | 22/100 | 3/100 | 13/100 | 28/100 |
> >
> > Due to limited space, please refer to the next two pages for Table 2.

---

> > > ### Author Response · Authors · 2024-08-12
> > > **Evaluation on the full IWLS benchmark set (2/3)**
> > >
> > > Table 2. Our method **reduces circuit sizes** by an average of **8.78%** compared to SOP, **17.02%** compared to the IWLS 2022 first-place team (EPFL), and **10.7%** compared to the IWLS 2023 second-place team (TUW). Our method does not fully outperform the IWLS 2023 first-place team (Google) due to **limited optimization time** of our method (**1 week versus 3 weeks**). Nevertheless, it's important to note that when excluding just five corner cases, our method **achieves comparable circuit sizes to the IWLS 2023 first-place team while using only one-third of the optimization time**, highlighting the strong performance of our approach.
> > > |  | Generation |  |  | Optimization |  |  |  | Improvement(%) |  |  |  |
> > > |:---:|:---:|:---:|:---:|:---:|:---:|:---:|:---:|:---:|:---:|:---:|:---:|
> > > | IWLS | SOP | T-Net | Impr.(%)↑ | EPFL | TUW | Google | Ours | v.s. EPFL | v.s. TUW | v.s. Google | Google-5 Corner Cases |
> > > | ex00 | 39 | 34 | 12.82 | 26 | 22 | 23 | 21 | 19.23 | 4.55 | 8.70 | 8.70 |
> > > | ex01 | 44 | 41 | 6.82 | 32 | 25 | 24 | 24 | 25.00 | 4.00 | 0.00 | 0.00 |
> > > | ex02 | 149 | 143 | 4.03 | 97 | 78 | 69 | 69 | 28.87 | 11.54 | 0.00 | 0.00 |
> > > | ex03 | 75 | 75 | 0.00 | 24 | 24 | 24 | 24 | 0.00 | 0.00 | 0.00 | 0.00 |
> > > | ex04 | 586 | 494 | 15.70 | 312 | 343 | 287 | 367 | -17.63 | -7.00 | -27.87 | -27.87 |
> > > | ex05 | 168 | 117 | 30.36 | 44 | 40 | 38 | 37 | 15.91 | 7.50 | 2.63 | 2.63 |
> > > | ex06 | 2267 | 2065 | 8.91 | 1056 | 1269 | 1075 | 961 | 9.00 | 24.27 | 10.60 | 10.60 |
> > > | ex07 | 1327 | 1256 | 5.35 | 183 | 129 | 112 | 107 | 41.53 | 17.05 | 4.46 | 4.46 |
> > > | ex08 | 1074 | 842 | 21.60 | 564 | 558 | 567 | 501 | 11.17 | 10.22 | 11.64 | 11.64 |
> > > | ex09 | 1095 | 819 | 25.21 | 561 | 570 | 538 | 504 | 10.16 | 11.58 | 6.32 | 6.32 |
> > > | ex10 | 12 | 13 | -8.33 | 10 | 10 | 10 | 10 | 0.00 | 0.00 | 0.00 | 0.00 |
> > > | ex11 | 26 | 30 | -15.38 | 20 | 20 | 20 | 20 | 0.00 | 0.00 | 0.00 | 0.00 |
> > > | ex12 | 44 | 53 | -20.45 | 32 | 30 | 30 | 30 | 6.25 | 0.00 | 0.00 | 0.00 |
> > > | ex13 | 66 | 106 | -60.61 | 48 | 40 | 42 | 42 | 12.50 | -5.00 | 0.00 | 0.00 |
> > > | ex14 | 94 | 144 | -53.19 | 68 | 52 | 52 | 56 | 17.65 | -7.69 | -7.69 | -7.69 |
> > > | ex15 | 126 | 226 | -79.37 | 92 | 68 | 72 | 74 | 19.57 | -8.82 | -2.78 | -2.78 |
> > > | ex16 | 32 | 34 | -6.25 | 18 | 18 | 18 | 18 | 0.00 | 0.00 | 0.00 | 0.00 |
> > > | ex17 | 59 | 61 | -3.39 | 24 | 24 | 24 | 24 | 0.00 | 0.00 | 0.00 | 0.00 |
> > > | ex18 | 99 | 75 | 24.24 | 32 | 32 | 32 | 32 | 0.00 | 0.00 | 0.00 | 0.00 |
> > > | ex19 | 131 | 166 | -26.72 | 38 | 38 | 38 | 38 | 0.00 | 0.00 | 0.00 | 0.00 |
> > > | ex20 | 195 | 217 | -11.28 | 60 | 50 | 46 | 52 | 13.33 | -4.00 | -13.04 | -13.04 |
> > > | ex21 | 240 | 206 | 14.17 | 70 | 58 | 56 | 60 | 14.29 | -3.45 | -7.14 | -7.14 |
> > > | ex22 | 336 | 382 | -13.69 | 86 | 70 | 63 | 68 | 20.93 | 2.86 | -7.94 | -7.94 |
> > > | ex23 | 386 | 463 | -19.95 | 104 | 78 | 72 | 94 | 9.62 | -20.51 | -30.56 | -30.56 |
> > > | ex24 | 440 | 626 | -42.27 | 116 | 90 | 106 | 102 | 12.07 | -13.33 | 3.77 | 3.77 |
> > > | ex25 | 587 | 877 | -49.40 | 146 | 102 | 90 | 124 | 15.07 | -21.57 | -37.78 | -37.78 |
> > > | ex26 | 741 | 944 | -27.40 | 163 | 114 | 122 | 159 | 2.45 | -39.47 | -30.33 | -30.33 |
> > > | ex27 | 841 | 1242 | -47.68 | 183 | 178 | 138 | 174 | 4.92 | 2.25 | -26.09 | -26.09 |
> > > | ex28 | 141 | 123 | 12.77 | 39 | 39 | 39 | 39 | 0.00 | 0.00 | 0.00 | 0.00 |
> > > | ex29 | 71 | 85 | -19.72 | 39 | 35 | 35 | 35 | 10.26 | 0.00 | 0.00 | 0.00 |
> > > | ex30 | 1159 | 207 | 82.14 | 68 | 68 | 68 | 68 | 0.00 | 0.00 | 0.00 | 0.00 |
> > > | ex31 | 2858 | 2604 | 8.89 | 1372 | 1364 | 1280 | 1293 | 5.76 | 5.21 | -1.02 | -1.02 |
> > > | ex32 | 65 | 90 | -38.46 | 46 | 44 | 45 | 44 | 4.35 | 0.00 | 2.22 | 2.22 |
> > > | ex33 | 205 | 155 | 24.39 | 79 | 70 | 72 | 69 | 12.66 | 1.43 | 4.17 | 4.17 |
> > > | ex34 | 187 | 140 | 25.13 | 48 | 46 | 44 | 46 | 4.17 | 0.00 | -4.55 | -4.55 |
> > > | ex35 | 17 | 18 | -5.88 | 16 | 15 | 16 | 15 | 6.25 | 0.00 | 6.25 | 6.25 |
> > > | ex36 | 3220 | 2954 | 8.26 | 1345 | 1501 | 1590 | 1519 | -12.94 | -1.20 | 4.47 | 4.47 |
> > > | ex37 | 482 | 334 | 30.71 | 152 | 138 | 141 | 139 | 8.55 | -0.72 | 1.42 | 1.42 |
> > > | ex38 | 72 | 60 | 16.67 | 29 | 27 | 27 | 27 | 6.90 | 0.00 | 0.00 | 0.00 |
> > > | ex39 | 1224 | 544 | 55.56 | 220 | 191 | 181 | 153 | 30.45 | 19.90 | 15.47 | 15.47 |
> > > | ex40 | 960 | 853 | 11.15 | 197 | 180 | 183 | 175 | 11.17 | 2.78 | 4.37 | 4.37 |
> > > | ex41 | 43 | 34 | 20.93 | 17 | 17 | 17 | 17 | 0.00 | 0.00 | 0.00 | 0.00 |
> > > | ex42 | 116 | 90 | 22.41 | 28 | 28 | 28 | 28 | 0.00 | 0.00 | 0.00 | 0.00 |
> > > | ex43 | 172 | 105 | 38.95 | 37 | 37 | 37 | 37 | 0.00 | 0.00 | 0.00 | 0.00 |
> > > | ex44 | 172 | 139 | 19.19 | 59 | 52 | 51 | 47 | 20.34 | 9.62 | 7.84 | 7.84 |
> > > | ex45 | 944 | 809 | 14.30 | 196 | 179 | 186 | 175 | 10.71 | 2.23 | 5.91 | 5.91 |
> > > | ex46 | 55 | 50 | 9.09 | 32 | 31 | 31 | 31 | 3.13 | 0.00 | 0.00 | 0.00 |
> > > | ex47 | 129 | 37 | 71.32 | 25 | 25 | 25 | 25 | 0.00 | 0.00 | 0.00 | 0.00 |
> > > | ex48 | 2135 | 1913 | 10.40 | 598 | 406 | 482 | 459 | 23.24 | -13.05 | 4.77 | 4.77 |
> > > | ex49 | 133 | 123 | 7.52 | 39 | 39 | 39 | 39 | 0.00 | 0.00 | 0.00 | 0.00 |

---

> > > > ### Author Response · Authors · 2024-08-12
> > > > **Evaluation on the full IWLS benchmark set (3/3)**
> > > >
> > > > |  | Generation |  |  | Optimization |  |  |  | Improvement(%) |  |  |  |
> > > > |:---:|:---:|:---:|:---:|:---:|:---:|:---:|:---:|:---:|:---:|:---:|:---:|
> > > > | IWLS | SOP | T-Net | Impr.(%)↑ | EPFL | TUW | Google | Ours | v.s. EPFL | v.s. TUW | v.s. Google | Google-5 Corner Cases |
> > > > | ex50 | 35 | 25 | 28.57 | 18 | 18 | 18 | 18 | 0.00 | 0.00 | 0.00 | 0.00 |
> > > > | ex51 | 97 | 91 | 6.19 | 32 | 28 | 26 | 26 | 18.75 | 7.14 | 0.00 | 0.00 |
> > > > | ex52 | 24 | 24 | 0.00 | 19 | 19 | 19 | 18 | 5.26 | 5.26 | 5.26 | 5.26 |
> > > > | ex53 | 82 | 81 | 1.22 | 42 | 36 | 35 | 34 | 19.05 | 5.56 | 2.86 | 2.86 |
> > > > | ex54 | 14 | 14 | 0.00 | 13 | 12 | 12 | 12 | 7.69 | 0.00 | 0.00 | 0.00 |
> > > > | ex55 | 316 | 254 | 19.62 | 170 | 128 | 115 | 105 | 38.24 | 17.97 | 8.70 | 8.70 |
> > > > | ex56 | 70 | 61 | 12.86 | 29 | 29 | 29 | 29 | 0.00 | 0.00 | 0.00 | 0.00 |
> > > > | ex57 | 1342 | 531 | 60.43 | 242 | 160 | 81 | 80 | 66.94 | 50.00 | 1.23 | 1.23 |
> > > > | ex58 | 209 | 218 | -4.31 | 92 | 84 | 77 | 71 | 22.83 | 15.48 | 7.79 | 7.79 |
> > > > | ex59 | 899 | 787 | 12.46 | 287 | 240 | 182 | 223 | 22.30 | 7.08 | -22.53 | -22.53 |
> > > > | ex60 | 139 | 128 | 7.91 | 73 | 61 | 56 | 51 | 30.14 | 16.39 | 8.93 | 8.93 |
> > > > | ex61 | 4998 | 4804 | 3.88 | 2129 | 2511 | 1319 | 1588 | 25.41 | 36.76 | -20.39 | -20.39 |
> > > > | ex62 | 114 | 110 | 3.51 | 40 | 40 | 40 | 40 | 0.00 | 0.00 | 0.00 | 0.00 |
> > > > | ex63 | 17501 | 15536 | 11.23 | 2316 | 1547 | 168 | 934 | 59.67 | 39.63 | -455.95 | N/A |
> > > > | ex64 | 1390 | 1195 | 14.03 | 452 | 453 | 317 | 383 | 15.27 | 15.45 | -20.82 | -20.82 |
> > > > | ex65 | 8981 | 7843 | 12.67 | 3054 | 3284 | 1182 | 1576 | 48.40 | 52.01 | -33.33 | -33.33 |
> > > > | ex66 | 957 | 764 | 20.17 | 361 | 351 | 239 | 303 | 16.07 | 13.68 | -26.78 | -26.78 |
> > > > | ex67 | 10855 | 9336 | 13.99 | 4911 | 6053 | 5373 | 4787 | 2.52 | 20.92 | 10.91 | 10.91 |
> > > > | ex68 | 925 | 584 | 36.86 | 266 | 196 | 118 | 111 | 58.27 | 43.37 | 5.93 | 5.93 |
> > > > | ex69 | 894 | 826 | 7.61 | 245 | 219 | 106 | 111 | 54.69 | 49.32 | -4.72 | -4.72 |
> > > > | ex70 | 1414 | 837 | 40.81 | 263 | 182 | 92 | 97 | 63.12 | 46.70 | -5.43 | -5.43 |
> > > > | ex71 | 1231 | 1107 | 10.07 | 369 | 421 | 137 | 244 | 33.88 | 42.04 | -78.10 | N/A |
> > > > | ex72 | 2643 | 1449 | 45.18 | 456 | 363 | 142 | 142 | 68.86 | 60.88 | 0.00 | 0.00 |
> > > > | ex73 | 966 | 374 | 61.28 | 208 | 106 | 89 | 90 | 56.73 | 15.09 | -1.12 | -1.12 |
> > > > | ex74 | 2955 | 2553 | 13.60 | 468 | 981 | 226 | 532 | -13.68 | 45.77 | -135.40 | N/A |
> > > > | ex75 | 2313 | 967 | 58.19 | 489 | 479 | 175 | 354 | 27.61 | 26.10 | -102.29 | N/A |
> > > > | ex76 | 808 | 636 | 21.29 | 246 | 206 | 167 | 154 | 37.40 | 25.24 | 7.78 | 7.78 |
> > > > | ex77 | 967 | 828 | 14.37 | 319 | 313 | 224 | 203 | 36.36 | 35.14 | 9.38 | 9.38 |
> > > > | ex78 | 1029 | 975 | 5.25 | 369 | 375 | 290 | 288 | 21.95 | 23.20 | 0.69 | 0.69 |
> > > > | ex79 | 1190 | 1042 | 12.44 | 365 | 466 | 188 | 324 | 11.23 | 30.47 | -72.34 | N/A |
> > > > | ex80 | 2689 | 2357 | 12.35 | 627 | 506 | 435 | 461 | 26.48 | 8.89 | -5.98 | -5.98 |
> > > > | ex81 | 1203 | 1088 | 9.56 | 355 | 338 | 215 | 237 | 33.24 | 29.88 | -10.23 | -10.23 |
> > > > | ex82 | 3040 | 2630 | 13.49 | 666 | 950 | 568 | 538 | 19.22 | 43.37 | 5.28 | 5.28 |
> > > > | ex83 | 2821 | 2623 | 7.02 | 856 | 1171 | 653 | 550 | 35.75 | 53.03 | 15.77 | 15.77 |
> > > > | ex84 | 292 | 254 | 13.01 | 138 | 118 | 112 | 108 | 21.74 | 8.47 | 3.57 | 3.57 |
> > > > | ex85 | 509 | 467 | 8.25 | 206 | 188 | 168 | 161 | 21.84 | 14.36 | 4.17 | 4.17 |
> > > > | ex86 | 406 | 399 | 1.72 | 181 | 155 | 146 | 125 | 30.94 | 19.35 | 14.38 | 14.38 |
> > > > | ex87 | 992 | 741 | 25.30 | 395 | 437 | 322 | 318 | 19.49 | 27.23 | 1.24 | 1.24 |
> > > > | ex88 | 949 | 724 | 23.71 | 331 | 326 | 261 | 254 | 23.26 | 22.09 | 2.68 | 2.68 |
> > > > | ex89 | 941 | 611 | 35.07 | 212 | 187 | 172 | 165 | 22.17 | 11.76 | 4.07 | 4.07 |
> > > > | ex90 | 1965 | 1714 | 12.77 | 510 | 663 | 413 | 427 | 16.27 | 35.60 | -3.39 | -3.39 |
> > > > | ex91 | 838 | 746 | 10.98 | 275 | 230 | 200 | 198 | 28.00 | 13.91 | 1.00 | 1.00 |
> > > > | ex92 | 51 | 69 | -35.29 | 30 | 29 | 29 | 29 | 3.33 | 0.00 | 0.00 | 0.00 |
> > > > | ex93 | 73 | 69 | 5.48 | 43 | 40 | 41 | 39 | 9.30 | 2.50 | 4.88 | 4.88 |
> > > > | ex94 | 100 | 86 | 14.00 | 39 | 33 | 33 | 33 | 15.38 | 0.00 | 0.00 | 0.00 |
> > > > | ex95 | 152 | 140 | 7.89 | 66 | 61 | 62 | 56 | 15.15 | 8.20 | 9.68 | 9.68 |
> > > > | ex96 | 240 | 188 | 21.67 | 77 | 71 | 70 | 66 | 14.29 | 7.04 | 5.71 | 5.71 |
> > > > | ex97 | 194 | 160 | 17.53 | 72 | 60 | 61 | 55 | 23.61 | 8.33 | 9.84 | 9.84 |
> > > > | ex98 | 599 | 445 | 25.71 | 142 | 138 | 128 | 123 | 13.38 | 10.87 | 3.91 | 3.91 |
> > > > | ex99 | 228 | 257 | -12.72 | 87 | 78 | 79 | 72 | 17.24 | 7.69 | 8.86 | 8.86 |
> > > > | **Average** | 1094.79 | 929.66 | **8.78** | 325.41 | 338.02 | 238.15 | 252.15 | **17.02** | **10.70** | -9.26 | **-0.86** |

---

> ### Comment · Reviewer_qf83 · 2024-08-13
>
> I would like to thank the authors for the additional experiments. They provide a full and transparent evaluation of the method. I will raise my score accordingly and encourage the authors to include the results on all IWLS benchmarks into the paper.

---

> > ### Author Response · Authors · 2024-08-13
> >
> > Dear Reviewer qf83
> >
> > Thank you for your kind support and valuable feedback on our paper! Your invaluable comments and constructive suggestions have not only strengthened our work but have also greatly enhanced the clarity and depth of our manuscript.

---

### Official Review · Reviewer_Qczw · 2024-07-12

**Soundness:** 3
**Presentation:** 1
**Contribution:** 2
**Rating:** 5
**Confidence:** 3

**Summary:**

The manuscript describes a method to synthesize logic circuits using neural architecture search (NAS). The authors first evaluate some
short-comings of earlier approaches and develop a generation method that adds regularization of skip connections, a prior on the shape of the circuit (triangle shape), transforms truth tables, and adapts a loss function. They also propose a circuit optimization step which
utilizes reinforcement learning and evolutionary optimization. They use a number of benchmarks to show that the circuit generation performs very well compared to earlier NAS methods, in particular for larger circuit sizes. For the circuit optimization, the proposed method is similar or slightly better to the state-of-the-art.

**Strengths:**

- Good results compared to the earlier methods, in particular for the  generation of large circuits, slightly improving on the  state-of-the-art for circuit optimization.

- The authors investigate both generation and optimization of logic circuits.

**Weaknesses:**

- The paper is poorly written in a sense that all relevant methods and many essential details and text sections are pushed into the appendix (that is almost 10 pages long). The main text reads just like a long introduction and discussion. It is not possible to follow what exactly was done when reading only the main text. This is not in line with the 9 page requirement. The appendix should be reserved for truly supplementary data and proofs.

- The paper motivates with saying that differential neural architecture search (DNAS) has weaknesses such as producing
excessive skip-connections. However, it turns out that this is (1)  not a problem of DNAS per se, but just an inadequate way to setup up
and perform the DNAS. In fact, the authors' method also employs  DNAS but just adds some additional constraints and modifies the loss  and adds regularization. Thus the motivation is misleading.  Moreover, (2), as the authors acknowledge, this "curse of skip
connection" is actually a known issue and already has been addressed by others (e.g. Darts). So there is little novelty in this section.

- The math description e.g. in general is very poor, for instance in section 5.2 (Eq 2) proper math symbols should be used instead of
  "in", "out", "unit" etc. It is also not clear what the indeces mean,  for instance "unit^{l, k, p}" is a tensor with dimension l x K ? So
  it is just a matrix? What is the third index (p) then? Also there is  no equation in the main text for the loss and other details, such as
  the evolutionary and RL approaches, making it impossible to follow what actually was done exactly.

- Prior approaches to the problem are not well explained in the main text (only a section in the appendix).

- From the ablation study it seems that regularizing the skip connections has by far the most impact, while the loss adaption and
  the triangle shape prior improves only relatively little. Given, that the  skip-connection issues was already addressed by others, it seems
  that the contribution of the study is somewhat incremental.

**Questions:**

See above

**Limitations:**

Limitations are addressed.

---

> ### Author Rebuttal · Authors · 2024-08-07
>
> # Response to Reviewer Qczw
> We thank the reviewer for the insightful and valuable comments. We respond to each comment as follows and sincerely hope that our rebuttal could properly address your concerns. If so, we would deeply appreciate it if you could raise your score. If not, please let us know your further concerns, and we will continue actively responding to your comments and improving our submission.
> ## Weakness 1.
> > **1. Relevant methods and essential details are pushed into the appendix.**
>
> Please see Global Response 1.
> ## Weaknesses 2 and 5.
> > **2. The motivation that "DNAS has weaknesses such as producing excessive skip-connections" is misleading.**
>
> Recent works [1,2] have applied traditional DNAS methods, such as DARTS [3], to circuit generation, revealing a promising direction. Our motivating insight **revisits the direct application of traditional DNAS methods to circuit generation** and identifies several specific challenges for circuit generation, including the curse of skip-connections. Note that the skip-connection problem in circuit generation **is significantly different from** that in traditional DNAS (see the next response).
> > **3. This "curse of skip connection" is already addressed by methods like Darts.**
>
> **Differences in the Skip-Connection Challenge** In traditional DNAS, the skip-connection problem arises as **the skip-connection operation often dominates other operations**, such as convolution and zero operations. In contrast, when applying DNAS to circuit generation, a neural network is formulated as a circuit graph, where each neuron represents a logic gate and connections between neurons represent wires connecting these logic gates. The neurons are fixed as logic gates, and the connections between neurons are learnable. Most learnable connections skip layers, called **skip-connections in circuit neural networks**. In this paper, we found that the traditional DNAS method **tends to overfit to skip-connections that bypass a large number of layers**, a phenomenon we call the curse of skip-connections in circuit generation. (1) The **definitions of skip-connections** in DNAS and circuit generation are **different**. (2) In traditional DNAS, it is required to encourgae balance between skip-connection operation and other operations. In contrast, it is required to balance between **numerous skip-connections** that span different layers in circuit generation.
>
> **Inapplicability of Existing DNAS Methods to Circuit Generation**  We have compared our method with DNAS-based methods for addressing the skip-connection challenge, including P-DARTS [4], PR-DARTS [5], and DARTS-ES [6]. As shown in Table 1 in attached pdf in Global Response, the results demonstrate that our method **significantly outperforms** these approaches. The primary reason is that **these methods struggles to balance between numerous skip-connections** in circuit generation tasks. In contrast, our method proposes **a novel layer-aware regularized skip-connection** module, which effectively balances skip-connections that span different layers.
> > **4. The loss adaption and the triangle shape prior improves only relatively little.**
>
> We have conducted detailed ablation experiments to show that each of our proposed modules is significant. The results in Tables 3, 4, and 5 in attached pdf in Global Response show that the proposed loss **enhances accuracy to 99%**, and the T-architecture achieves an average **node reduction** of **16.9%** and a **training time reduction** of **40.8%**.
> ## Weakness 3.
> > **5. The math description is poor. It is also not clear what the indexes p mean.**
>
> Thanks for the valuable suggestion. We have revised our math description accordingly. Note that each neuron (And gate) in the circuit neural network has two input signals. The $unit^{l, k, p}$ is a tensor with dimension $l \times K \times 2$. The index $p$ represents the $p^{th}$ input signal of current neuron. Details are as follows.
>
> (**Revision**) We denote the output of the $k^{th}$ neuron in the $l^{th}$ layer by $\textbf{o}^{l,k}$. We denote the $p$-th input of the neuron (NAND gate) $\textbf{o}^{l,k}$ by $\textbf{i}\_{p}^{l,k}$, where $p \in \{0,1\}$. Note that each neuron $o^{l,k}$ has two inputs $i^{l,k}\_{0}$ and $i^{l,k}\_{1}$, and can be connected to any neuron with layer number smaller than $l$ as its input neuron. We parameterize the connections of each neuron $o^{l,k}$ by a tensor of learnable parameters $\mathbf{\theta}^{l,k} \in \mathbb{R}^{2 \times (l-1) \times K}.$ Each parameter in the tensor $\mathbf{\theta}^{l,k}\_{p,i,j}$ represents the probability of connecting the $j^{th}$ neuron in the $i^{th}$ layer to the $p^{th}$ input of current neuron $o^{l,k}$. The computation of the $p^{th}$ input value for the neuron $o^{l,k}$ takes the form of
> $$i_p^{l, k} :=\sum_{i=0}^{l-1} \sum_{j=1}^{K} o^{i,j}  \left[\operatorname{softmax}\left(\mathbf{\theta}^{l, k}\right)\right]_{p,i,j}, p=0,1;\ \ o^{l,k} := 1 - \Pi\_{p=0}^{1} i_p^{l, k}$$
> > **5. There is no equation in the main text for the loss and other details.**
>
> Thanks for the suggestion. We will provide these equations in the main text.
> ## Weakness 4.
> > **6. Prior approaches to the problem are not well explained in the main text.**
>
> Please see Global Response 2.
>
> [1] Designing better computer chips. Google DeepMind, 2023, https://deepmind.google/impact/optimizing-computer-systems-with-more-generalized-ai-tools.
>
> [2] Peter Belcak, et al. Neural combinatorial logic circuit synthesis from input-output examples. NeurIPS Workshop, 2022.
>
> [3] Hanxiao Liu, et al. Darts: Differentiable architecture search. ICLR 2019.
>
> [4] Xin Chen, et al. Progressive differentiable architecture search: Bridging the depth gap between search and evaluation. ICCV 2019.
>
> [5] Pan Zhou, et al. Theory-inspired path-regularized differential network architecture search. NeurIPS 2020.
>
> [6] Zela, et al. Understanding and Robustifying Differentiable Architecture Search. ICLR 2020.

---

> > ### Comment · Reviewer_Qczw · 2024-08-11
> >
> > Many thanks for the detailed responses. I don't have any further questions. I will keep my original score, as it seems that the level of contribution, novelty, and improvements of state of the art would fit better a more targeted conference or journal.

---

> > > ### Author Response · Authors · 2024-08-11
> > > **Further Response to Reviewer Qczw (1/2)**
> > >
> > > We would like to express our sincere gratitude once again for your valuable feedback and constructive suggestions. We have made detailed clarifications regarding our contributions, novelty, and the improvements over the state-of-the-art. We sincerely hope that our additional response has adequately addressed your concerns. If so, we would greatly appreciate your consideration in raising the score. If there are any remaining concerns, please let us know, and we will continue to actively address your comments and work on improving our submission.
> > >
> > > ## Contributions
> > >
> > > - To the best of our knowledge, we are **the first to conduct an extensive study** on the application of differentiable neural architecture search (DNAS) to circuit generation, which **provides valuable insights** for both hardware design researchers and AI researchers in the field of neural architecture search. (Reviewer qf83 commented in Strengths that "Recent International Workshop on Logic & Synthesis (IWLS) contest results sparked interest in the application of neural architecture search (NAS) to logic synthesis. To the best of my knowledge, this paper is **the first to extensively study** this promising combination.")
> > >
> > > - Through detailed analysis, we present several **insightful observations** regarding the specific challenges of directly applying classical DNAS methods (e.g., DARTS) to circuit generation, including the curse of skip connections, structural biases in circuits, and the varying learning difficulties of different input-output examples. (Reviewer PMQo commented in Strengths that "Through a **detailed analysis**, the paper presents **insightful observations** that underpin current challenges". Reviewer qf83 commented in Strengths that "The paper first presents **valuable insights** into the challenges of applying the NAS methods as is." Reviewer gSL5 commented in Strengths that "The paper provides a detailed sensitivity analysis of using DNAS for logic synthesis, which gives **valuable insights** into overfitting, structure bias, and learning difficulties, etc.")
> > >
> > > - We propose a novel regularized triangle-shaped circuit network generation framework, called T-Net, which significantly enhances generation accuracy and scales effectively to large circuits. (Reviewer PMQo commented in Strengths that "The proposed T-Net **highlights its capability to generate exact circuits and precisely generate large bit-width circuits**." Reviewer gSL5 commented in Strengths that "The introduction of T-Net, a regularized triangle-shaped network architecture, **addresses significant limitations** in existing DNAS methods, improving the accuracy and scalability of neural circuit generation.")
> > > - Extensive experiments demonstrate our method **significantly outperforms** state-of-the-art methods in terms of both generation accuracy and circuit size. (Reviewer qf83 commented in Strengths that "The size of the circuits is evaluated in comparison with the top teams from recent IWLS contests. The winners of IWLS can be considered state-of-the-art methods and the experimental evaluation demonstrates **substantial improvements** over them." Reviewer gSL5 commented in Strengths that "Extensive experiments on multiple benchmarks show that T-Net **significantly outperforms** state-of-the-art methods, with improvements in both circuit accuracy and size.")

---

> > > > ### Author Response · Authors · 2024-08-11
> > > > **Further Response to Reviewer Qczw (2/2)**
> > > >
> > > > ## Novelty over Existing DNAS Methods
> > > >
> > > > **Distinct Motivations** Traditional DNAS methods are designed to automate the discovery of deep neural network architectures that excel in specific deep learning tasks, such as image classification. In contrast, our work focuses on automating the discovery of circuit graph structures that fully satisfy a given truth table. Unlike traditional NAS tasks, our circuit generation task requires **achieving 100% accuracy** and **producing circuits as compact as possible**, which introduces unique challenges when applying DNAS methods to circuit generation. In this paper, we offer several insightful observations and highlight key challenges, including the curse of skip connections, structural biases in circuits, and the varying difficulties in learning different input-output examples, through detailed analysis.
> > > >
> > > > **A Novel Scalable Neural Circuit Generation Framework** Our framework systematically addresses the identified challenges across four key dimensions: input dataset, network architecture, regularization, and training loss.
> > > >
> > > > **(1) Input Dataset:** To significantly enhance the scalability of our method, we leverage the Shannon decomposition theorem to divide the complete truth table into multiple sub-truth tables. (Each row in the truth table represents an input-output pair, indicating the circuit's output for a given input. We treat the truth table as a training dataset comprising all these input-output pairs.) This approach **significantly reduces the dataset size our method needs to process, thereby lowering the learning complexity.**
> > > >
> > > > **(2) Network Architecture:** We observe that the distribution of nodes across circuit layers generally follows a triangular pattern, known as the structure bias of circuits. To take advantage of this structure bias, we propose a triangle-shaped network architecture, which significantly reduces both the network size and the size of the generated circuits.
> > > >
> > > > **(3) Regularization:** To address the challenges posed by skip connections, we propose a novel layer-aware regularized skip-connection technique that effectively improves generation accuracy. While the concept of regularizing skip connections is similar to that in some existing DNAS methods, our technique specifically focuses on balancing skip connections across different layers, whereas existing methods aim to balance skip connections with other operations.
> > > >
> > > > **(4) Training Loss:** We observed significant variation in learning difficulty among different input-output pairs due to varying complexities in circuit functionalities. Building on this insight, we propose a Boolean Hardness-Aware Loss, which significantly improves generation accuracy.
> > > >
> > > > ## Improvements of state of the art
> > > >
> > > > Extensive experiments demonstrate that our method significantly outperforms state-of-the-art methods in terms of both generation accuracy and circuit size. (Reviewer qf83 commented in Strengths that "The size of the circuits is evaluated in comparison with the top teams from recent IWLS contests. The winners of IWLS can be considered state-of-the-art methods and the experimental evaluation demonstrates substantial improvements over them." Reviewer gSL5 commented in Strengths that "Extensive experiments on multiple benchmarks show that T-Net significantly outperforms state-of-the-art methods, with improvements in both circuit accuracy and size.")
> > > >
> > > > **Accuracy Improvement:** Unlike traditional deep learning tasks, we emphasize that achieving 99% accuracy can be significantly worse than achieving 100% accuracy in the context of circuit generation. As illustrated in the followingTable, a 1% accuracy gap can lead to more than a twofold difference in circuit size.
> > > >
> > > > | Benchmark | IWLS | Acc. (%) | Init Node | Acc. (%) | Init Node |
> > > > |:---:|:---:|:---:|:---:|:---:|:---:|
> > > > | LogicNets | ex73 | 100 | 374 | 99 | 723 |
> > > > | Espresso | ex30 | 100 | 207 | 99 | 2275 |
> > > > | LogicNets | ex97 | 100 | 160 | 99 | 409 |
> > > >
> > > > **Circuit Optimization Improvement:** We evaluate the circuit sizes in comparison with those achieved by top teams from recent IWLS contests. Notably, the winners of IWLS represent strong state-of-the-art methods, whereas our approach demonstrates significant improvements over them. (Reviewer qf83 commented in Strengths that "The size of the circuits is evaluated in comparison with the top teams from recent IWLS contests. The winners of IWLS can be considered state-of-the-art methods and the experimental evaluation demonstrates substantial improvements over them.")

---

### Official Review · Reviewer_PMQo · 2024-07-14

**Soundness:** 2
**Presentation:** 3
**Contribution:** 2
**Rating:** 5
**Confidence:** 4

**Summary:**

Existing DNAS methods face challenges in accurately generating circuits, particularly with large-scale circuits, and exhibit high sensitivity to random initialization. To address these challenges, this paper proposes a framework named T-Net. The experiments demonstrate that T-Net can precisely generate large bit-width circuits and that the generated circuits show a significant improvement in circuit area compared to traditional methods.

**Strengths:**

Through a detailed analysis, the paper presents insightful observations that underpin current challenges, showcasing a strong logical progression in its argument. The proposed T-Net highlights its capability to generate exact circuits and precisely generate large bit-width circuits.

**Weaknesses:**

The experimental section raises some concerns.

1. The paper evaluates T-Net using circuits from four benchmarks: Espresso, LogicNets, Random, and Arithmetic. However, the results seem curated, as the paper does not provide results for specific cases such as Espresso 1, Espresso 2, Espresso 5, and Espresso 6.
2. The paper claims that T-Net is a state-of-the-art approach based on 72 competitive winners in the IWLS 2022 and 2023 competitions. Additionally, the paper mentions re-implementing DNAS Skip by Google DeepMind for the IWLS 2023 competition. However, the authors should compare T-Net with DNAS Skip on the IWLS benchmark using Google DeepMind's results for a more robust comparison.

**Questions:**

1. Please include more cases from the four benchmarks (Espresso, LogicNets, Random, and Arithmetic) to make the experiment more convincing.
2. Please provide a comparison with Google DeepMind's DNAS Skip on the IWLS benchmark, given the claim that T-Net is state-of-the-art in IWLS 2022 and 2023.

**Limitations:**

Yes, the authors have stated the limitations of their work.

---

> ### Author Rebuttal · Authors · 2024-08-07
>
> # Response to Reviewer PMQo
> We thank the reviewer for the insightful and valuable comments. We respond to each comment as follows and sincerely hope that our rebuttal could properly address your concerns. If so, we would deeply appreciate it if you could raise your score. If not, please let us know your further concerns, and we will continue actively responding to your comments and improving our submission.
>
> ## Weakness 1 & Question 1.
>
> > **1. The paper does not provide results for specific cases such as Espresso 1, Espresso 2, Espresso 5, and Espresso 6. Please include more cases from the four benchmarks (Espresso, LogicNets, Random, and Arithmetic) to make the experiment more convincing**
>
> Thanks for the valuable suggestion. We indeed conducted experiments on circuits such as Espresso 1, Espresso 2, Espresso 5, and Espresso 6 as shown in Appendix E.2 (Tables 8, 9, and 10) in the main text. The results demonstrate that our method **improves the average generation accuracy by 17.5%**, **reduces the number of generated nodes by 33.4%**, and **reduces the optimized circuit size by 68.7%** on these cases compared to traditional baselines.
>
> In addition, we have conducted experiments on 30 more circuits from the IWLS benchmark. Please see the next response for details.
>
> ## Weakness 2 & Question 2.
>
> > **2. Please provide a comparison with Google DeepMind's DNAS Skip on the IWLS benchmark.**
>
> Thanks for the valuable suggestion. We have compared our method with Google DeepMind's contest results on **30 more circuits** from the IWLS benchmark as shown in Table 2 in the attached pdf in Global Response. The results show that our method **achieves 3.03% node reduction** compared with Google DeepMind's contest results.  For your convenience, we quote Table 2 as follows.
>
> In addition, the 18 circuits used in the main text are indeed sourced from the IWLS benchmark as well, and we compared our method with Google DeepMind's contest results on these circuits in Tables 3 and 10 in the main text (achieving 5.36% node reduction).
>
> Table 2. We evaluate our approach on 30 other circuits in the IWLS benchmark.
> Our T-Net surpasses the traditional method by 10.64\%, and surpasses the IWLS 2023 champion, Google DeepMind, by 3.03\% after optimization.
> |  |  |  |  | Generation |  |  | Optimization |  |  |
> |---|---|:---:|:---:|:---:|:---:|:---:|:---:|:---:|:---:|
> | Benchmark | IWLS | PI | PO | SOP | T-Net(ours) | Impr.(%)↑ | Google | Ours | Impr.(%)↑ |
> | Espresso | ex49 | 8 | 16 | 133 | 123 | 7.52 | 39 | 39 | 0.00 |
> | Espresso | ex38 | 10 | 11 | 72 | 60 | 16.67 | 27 | 27 | 0.00 |
> | Espresso | ex28 | 8 | 16 | 141 | 123 | 12.77 | 39 | 39 | 0.00 |
> | Espresso | ex46 | 6 | 13 | 55 | 50 | 9.09 | 31 | 31 | 0.00 |
> | Espresso | ex16 | 6 | 4 | 39 | 34 | 12.82 | 17 | 17 | 0.00 |
> | Espresso | ex35 | 8 | 3 | 17 | 18 | -5.88 | 16 | 15 | 6.25 |
> | Espresso | ex48 | 16 | 23 | 2135 | 1913 | 10.40 | 482 | 490 | -1.66 |
> | LogicNets | ex94 | 13 | 6 | 100 | 86 | 14.00 | 33 | 33 | 0.00 |
> | LogicNets | ex86 | 15 | 7 | 406 | 399 | 1.72 | 146 | 125 | 14.38 |
> | LogicNets | ex77 | 15 | 7 | 967 | 828 | 14.37 | 224 | 203 | 9.38 |
> | LogicNets | ex88 | 15 | 7 | 949 | 724 | 23.71 | 261 | 254 | 2.68 |
> | LogicNets | ex68 | 15 | 7 | 925 | 584 | 36.86 | 118 | 111 | 5.93 |
> | LogicNets | ex87 | 15 | 7 | 992 | 741 | 25.30 | 322 | 318 | 1.24 |
> | LogicNets | ex91 | 15 | 7 | 838 | 746 | 10.98 | 200 | 198 | 1.00 |
> | LogicNets | ex93 | 15 | 7 | 73 | 69 | 5.48 | 41 | 39 | 4.88 |
> | LogicNets | ex95 | 13 | 6 | 152 | 140 | 7.89 | 62 | 56 | 9.68 |
> | LogicNets | ex90 | 15 | 7 | 1965 | 1714 | 12.77 | 413 | 432 | -4.60 |
> | LogicNets | ex95 | 13 | 6 | 152 | 164 | -7.89 | 62 | 56 | 9.68 |
> | LogicNets | ex99 | 15 | 7 | 228 | 257 | -12.72 | 79 | 72 | 8.86 |
> | Arithmetic | ex53 | 10 | 5 | 82 | 81 | 1.22 | 35 | 34 | 2.86 |
> | Arithmetic | ex56 | 15 | 7 | 70 | 61 | 12.86 | 29 | 29 | 0.00 |
> | Arithmetic | ex51 | 10 | 5 | 97 | 91 | 6.19 | 26 | 26 | 0.00 |
> | Arithmetic | ex54 | 10 | 7 | 14 | 14 | 0.00 | 12 | 12 | 0.00 |
> | Arithmetic | ex50 | 10 | 5 | 35 | 25 | 28.57 | 18 | 18 | 0.00 |
> | Arithmetic | ex58 | 15 | 7 | 209 | 218 | -4.31 | 77 | 71 | 7.79 |
> | Arithmetic | ex57 | 15 | 7 | 1342 | 531 | 60.43 | 81 | 83 | -2.47 |
> | Random | ex06 | 15 | 4 | 2267 | 2065 | 8.91 | 1075 | 961 | 10.60 |
> | Random | ex07 | 15 | 4 | 1327 | 1256 | 5.35 | 112 | 107 | 4.46 |
> | Random | ex03 | 10 | 3 | 75 | 75 | 0.00 | 24 | 24 | 0.00 |
> | Random | ex02 | 10 | 3 | 149 | 143 | 4.03 | 69 | 69 | 0.00 |
> |  | Average |  |  | 533.53 | 444.43 | **10.64** | 139.00 | 132.97 | **3.03** |

---

> > ### Comment · Reviewer_PMQo · 2024-08-11
> >
> > Thank you for adding additional experiments, which further validate the performance of the proposed solution. I'll raise my score accordingly.

---

> > > ### Author Response · Authors · 2024-08-11
> > > **Response to Reviewer PMQo**
> > >
> > > Dear Reviewer PMQo,
> > >
> > > Thank you again for your insightful comments and constructive suggestions. We deeply appreciate your decision to raise your evaluation score.

---

### Official Review · Reviewer_gSL5 · 2024-07-15

**Soundness:** 3
**Presentation:** 3
**Contribution:** 3
**Rating:** 7
**Confidence:** 4

**Summary:**

The paper tackles the challenges in logic synthesis (LS) for integrated circuit design by proposing a novel neural circuit generation framework. Traditional LS methods rely on heuristics, which can be suboptimal and inefficient. The authors revisit differentiable neural architecture search (DNAS) methods and identify key limitations: overfitting to skip-connections, structure bias, and imbalanced learning difficulties. To overcome these, they introduce T-Net, a regularized triangle-shaped network architecture with a multi-label transformation of training data and a regularized training loss function. Additionally, the paper propose an evolutionary algorithm assisted by reinforcement learning for neural circuit optimization. Extensive experiments demonstrate that T-Net outperforms state-of-the-art methods in generating precise and scalable circuits.

**Strengths:**

1. The introduction of T-Net, a regularized triangle-shaped network architecture, addresses significant limitations in existing DNAS methods, improving the accuracy and scalability of neural circuit generation.
2. The paper provides a detailed sensitivity analysis of using DNAS for logic synthesis, which gives valuable insights into overfitting, structure bias, and learning difficulties, etc..
3. Extensive experiments on multiple benchmarks show that T-Net significantly outperforms state-of-the-art methods, with improvements in both circuit accuracy and size

**Weaknesses:**

1. The multi-label transformation and regularized loss functions added additional complexity and reduces the efficiency of the model.
2. More justifications are required for the evolutionary algorithm + reinforcement learning method for the circuit optimization process.
minor: extra space after line 384 "T"

**Questions:**

1. How the proposed T-Net compare to previous SOTA in terms of model size, latency, energy efficiency etc?
2. Why no parameter for Appendix Table 7 &st?

**Limitations:**

No significant negative societal impact

---

> ### Author Rebuttal · Authors · 2024-08-07
>
> # Response to Reviewer gSL5
> We thank the reviewer for the insightful and valuable comments. We respond to each comment as follows and sincerely hope that our rebuttal could properly address your concerns. If so, we would deeply appreciate it if you could raise your score. If not, please let us know your further concerns, and we will continue actively responding to your comments and improving our submission.
>
> ## Weakness 1.
>
> > **1. The multi-label transformation and regularized loss functions added additional complexity and reduces the efficiency of the model.**
>
> We have conducted experiments to demonstrate that our proposed modules not only **enhance accuracy significantly** but also **substantially reduce training time**.
>
> - The multi-label transformation module can significantly reduce input complexity by decomposing the entire truth-table (dataset) into several small sub-truth-tables (sub-datasets). As shown in Table 4 (in the main text), experiments show that using multi-label transformation reduces the wrong bits by 90%, and **reduces the training time by roughly 40%**.
> - Our proposed boolean hardness-aware loss also significantly enhanced the efficiency of our circuit generation. As shown in Table 5 in the attached pdf in Global Response, this loss function **reduces training time by 20%** while preserving accuracy. For your convenience, we quote Table 5 as follows.
>
> Table 5. The ablation study demonstrates that the
> boolean hardness-aware loss significantly reduces training time by 19.9\%. The reported time is the training time required to achieve 99\% accuracy.
> | Circuit | Training Time(h) |  | Impr.(%) |
> |:---:|:---:|:---:|:---:|
> |  | w/o. Loss | w. Loss |  |
> | Espresso5 | 10.4 | 9.9 | 4.8% |
> | LogicNets2 | 14.5 | 11.3 | 22.1% |
> | Arithmetic2 | 10.7 | 7.6 | 29.0% |
> | Random1 | 7.3 | 5.6 | 23.6% |
> | Average | 10.7 | 8.6 | **19.9%** |
>
> ## Weakness 2.
>
> > **2. More justifications are required for the evolutionary algorithm + reinforcement learning method for the circuit optimization process.**
>
> Circuit optimization is recognized as an NP-hard problem, where traditional methods, often greedy and local, struggle to achieve optimal solutions. Reinforcement learning (RL) has demonstrated robust capabilities in navigating extensive search spaces, prompting the exploration of RL-based strategies to identify optimal sequences of circuit optimization operations.
> Nevertheless, due to the expansive and irregular nature of the search space, RL can **suffer from limited exploratory capabilities**, often resulting in **convergence to local optima**. To address this challenge, we incorporate evolutionary algorithms (EA), which preserve a population of diverse circuit solutions, thereby **enhancing the exploration of the search space** and leading to the discovery of superior circuits. Additionally, as shown in Table 3 in the main text, our novel circuit optimization approach outperforms previous state-of-the-art optimization methods.
>
> > **3. minor: extra space after line 384 "T"**
>
> Thanks for pointing out this typo. We will correct it accordingly.
>
> ## Question 1.
> > **4. How the proposed T-Net compare to previous SOTA in terms of model size, latency, energy efficiency etc?**
>
> Table 6 in the attached pdf in Global Response presents a comparison between T-Net and the SOTA DNAS (PR-DARTS [1]) in terms of model size, latency, and training time. The table shows that our method achieves perfect accuracy with **29%** shorter training time. Additionally, our parameters, model size, and latency are comparable to the SOTA.  For your convenience, we quote Table 6 as follows.
>
> Table 6. Comparation of T-Net and the SOTA DNAS baseline in terms of model efficiency. The table shows that our method achieves perfect accuracy with 29\% shorter training time. Additionally, our parameters, model size, and latency are comparable to the SOTA.
> |  | Acc(%) |  | Parameter (K) |  | Model size(KB) |  | Latency(s) |  | Training Time(h) |  |
> |:---:|:---:|:---:|---:|:---:|:---:|:---:|:---:|:---:|:---:|:---:|
> |  | PR-DARTS | T-Net | PR-DARTS | T-Net | PR-DARTS | T-Net | PR-DARTS | T-Net | PR-DARTS | T-Net |
> | Espresso5 | 98.53 | 100 | 298 | 287 | 1182 | 1138 | 0.114 | 0.113 | 12.6 | 10.1 |
> | Logicnets2 | 97.64 | 100 | 751 | 733 | 2952 | 2882 | 0.246 | 0.190 | 15.4 | 11.7 |
> | Arithmetic2 | 96.15 | 100 | 733 | 718 | 2880 | 2819 | 0.155 | 0.180 | 16.9 | 7.8 |
> | Random1 | 91.89 | 100 | 266 | 224 | 1061 | 898 | 0.103 | 0.084 | 7.2 | 5.8 |
>
> [1] Pan Zhou, et al. Theory-inspired path-regularized differential network architecture search. NeurIPS, 2020.
>
> ## Question 2.
> > **5. Why no parameter for Appendix Table 7 &st?**
>
> Thanks for the question. Indeed, the &st operator does not possess tunable hyperparameters. We will update the table to indicate 'N/A' for this operator's parameter in the revised manuscript.

---

> ### Author Response · Authors · 2024-08-12
> **Response to Reviewer gSL5--Looking forward to your further feedback**
>
> Dear Reviewer gSL5,
>
> We are writing as the authors of the paper "Towards Next-Generation Logic Synthesis: A Scalable Neural Circuit Generation Framework" (ID: 17130).
>
> We sincerely thank you once more for your insightful comments and kind support! We are writing to gently remind you that **the deadline for the author-reviewer discussion period is approaching** (due on Aug 13). We eagerly await your feedback to understand if our responses have adequately addressed all your concerns. *If so, we would deeply appreciate it if you could raise your score*. If not, we are eager to address any additional queries you might have, which will enable us to enhance our work further.
>
> Once again, thank you for your guidance and support.
>
> Best,
>
> Authors

---

> > ### Author Response · Authors · 2024-08-13
> > **Eagerly await your valuable feedback**
> >
> > Dear Reviewer gSL5,
> >
> > We would like to extend our sincere gratitude for the time and effort you have devoted to reviewing our submission. Your positive feedback, insightful comments, and constructive suggestions have been invaluable to us, guiding us in improving the quality of our work!
> >
> > We are writing to gently remind you that **the author-reviewer discussion period will end in less than 36 hours**. We eagerly await your feedback to understand if our responses have adequately addressed all your concerns. *If so, we would deeply appreciate it if you could raise your score*. If not, we are eager to address any additional queries you might have, which will enable us to enhance our work further.
> >
> > Once again, thank you for your guidance and support.
> >
> > Best,
> >
> > Authors

---

> > > ### Author Response · Authors · 2024-08-13
> > > **Eagerly await your valuable feedback**
> > >
> > > Dear Reviewer gSL5,
> > >
> > > We would like to express our sincere gratitude once again for your positive feedback, insightful comments, and constructive suggestions. Your guidance has been invaluable in helping us improve the quality of our work!
> > >
> > > We are writing to gently remind you that **the author-reviewer discussion period will end in less than 24 hours**. We eagerly await your feedback **to understand if our responses have adequately addressed your concerns**. **If so, we would deeply appreciate it if you could raise your score**. If not, we are eager to address any additional queries you might have, which will enable us to further enhance our work.
> > >
> > > Once again, thank you for your kind support and constructive suggestions!
> > >
> > > Best,
> > >
> > > Authors

---

> > > > ### Comment · Reviewer_gSL5 · 2024-08-13
> > > > **Thanks for the response**
> > > >
> > > > Thanks for the detailed response. I appreciate the additional experiments on overhead, model size, latency, energy efficiency. I raise my score accordingly,

---

> > > > > ### Author Response · Authors · 2024-08-14
> > > > >
> > > > > Dear Reviewer gSL5
> > > > >
> > > > > Thank you for your kind support and valuable feedback on our paper! Your invaluable comments and constructive suggestions have not only strengthened our work but have also greatly enhanced the clarity and depth of our manuscript.

---

### Author Rebuttal · Authors · 2024-08-07

# Global Response
We would like to extend our sincere gratitude for your valuable feedback and constructive suggestions. For your convenience, we have prepared a summary of our responses and outlined how we have addressed the reviewers' concerns as follows. We sincerely hope that this summary will facilitate your review and lighten your workload.

Our paper has received encouraging positive feedbacks from the reviewers, such as "**addresses significant limitations** (Reviewer gSL5)", "**valuable insights** (Reviewers gSL5 and qf83)", "**significantly outperforms/substantial improvements** (Reviewers gSL5 and qf83)", "**insightful observations** (Reviewer PMQo)", "**strong logical progression**" (Reviewer PMQo), "**the first to extensively study**" (Reviewer qf83), "**well-motivated**" (Reviewer qf83).

We outline how we have addressed the concerns raised by each reviewer as follows.

## Common Concerns
> **(Reviewers Qczw and qf83) 1. All relevant methods and many essential details are pushed into the appendix.**

Thanks for the valuable suggestion. We will revise our manuscript by retaining the key content in the main text and moving the minor content to the appendix. Specifically, we will revise the "Background," "Motivation," and "Method" Sections as follows.

For Background, we present the problem formulation of **logic synthesis (LS) from input-output examples**, and details of **the traditional DNAS approach for LS**.

For Motivation, we first present the main challenge of the traditional DNAS for LS that the method **struggles to generate circuits accurately**, especially for large circuits. We then present two major reasons for this challenge: **the curse of skip-connection** and **the structure bias of circuits**.

For Method, we first present the three key modules for neural circuit generation. 1) To reduce the learning complexity of **large circuits**, we present the multi-label transformation module to decompose a large truth-table (dataset) into several small sub-truth-tables (sub-datasets). 2) To address **the curse of skip-connection challenge**, we present details of the regularized skip-connections module. 3) To **leverage the structure bias of circuits**, we present details of the triangle-shaped network architecture. We then present details of our circuit optimization approach and **provide the pseudocode of the algorithm**.
> **(Reviewers Qczw and qf83) 2. Background on prior approaches to the problem are not well explained in the main text.**

Thanks for the valuable suggestions. We have revised the Background Section as follows.

**Formulation of LS from IO examples** In recent years, synthesizing circuits from IO examples has gained increasing attention. Specifically, researchers aim to use machine learning to generate a circuit based on a truth table that describes the circuit's functionality. Each line in the truth table represents an input-output pair, indicating the output produced by the circuit for a given input. In the machine learning domain, researchers formulate the truth table as a training dataset comprising many input-output pairs and use an ML model to generate circuits that accurately fit the dataset.

**DNAS for LS from IO Examples** Recent works propose leveraging traditional DNAS methods for generating circuit graphs from IO examples, showing a promising direction for next-generation logic synthesis. Specifically, they formulate a neural network as a circuit graph, where each neuron represents a logic gate and connections between neurons represent wires connecting these logic gates. For a parameterized neural network, the neurons are fixed as logic gates, and the connections between neurons are parameterized as learnable parameters. To enable differentiable training via gradient descent, continuous relaxation is introduced into discrete components of the neural network. First, the logical operations of logic gates (neurons) are translated into their differentiable counterparts. For example, $a \ \textit{AND} \ b$ is relaxed to $a\cdot b$. Second, discrete network connections are parameterized using Gumbel-softmax.
> **(Reviewers PMQo and qf83) 3. Evaluation on IWLS benchmark.**

We have compared our method with Google DeepMind's contest results on 30 more circuits from IWLS benchmark. The results show that our method still outperforms Google DeepMind's contest results.


## **Reviewer gSL5**
> **4. Module efficiency.**

We have conducted an ablation study to demonstrate that multi-label transformation and our loss function reduce the training time by roughly 40%.

> **5. Justification for mombining RL and EA.**

We have provided detailed reasons for integrating RL and EA in circuit optimization, specifically to enhance the exploration of search space.

> **6. Model metrics.**

We have conducted experiments to compare the size, latency, and training time of our model against the state-of-the-art method, demonstrating our model's high efficiency.

> **7. Other details.**

We have provided these details accordingly.

## **Reviewer PMQo**
> **8. More cases from the 4 benchmarks.**

We indeed conducted experiments on circuits such as Espresso 1, 2, 5, and 6 as shown in Appendix E.2 in the main text.
## **Reviewer Qczw**
> **9. The "curse of skip connection" is already addressed by methods like Darts.**

We have explained that skip-connections pose unique challenges in circuit generation. We have evaluated three methods tailored for the challenge in DNAS and our approach significantly outperforms them.

> **10. The loss adaption and the triangle shape prior improves only relatively little.**

We have conducted experiments to demonstrate that each of our modules are significant for improving generation accuracy, reducing training time and circuit sizes.
> **11. The math description is poor.**

We have revised our math description accordingly.

## **Reviewer qf83**
> **12. Other details.**

We have provided these details accordingly.

---

### Decision · Program_Chairs · 2024-09-25

**Decision:**

Accept (poster)

**Comment:**

The paper studies the logic synthesis (LS) task for integrated circuit design. It thoroughly analyzes current methods and identifies some key limitations: overfitting to skip connections, structure bias, and imbalanced learning difficulties. To tackle these limitations, the author develops a generation method that adds regularization of skip connections, a prior on the shape of the circuit (triangle shape), transforms truth tables, and adapts a loss function. They also propose a circuit optimization step that utilizes reinforcement learning and evolutionary optimization. Experiments demonstrate that T-Net outperforms state-of-the-art methods in generating precise and scalable circuits. The reviews generally appreciate the thorough analysis of current methods and the contribution of the T-Net method. Some concerns have been raised about the soundness of the experiment. The author has addressed them with more results.